# Effect of iron oxide on nitrification in two agricultural soils with different pH

Xueru Huang[1], Xia Zhu-Barker[2], William R. Horwath[2], Sarwee J. Faeflen[1], Hongyan Luo[1], Xiaoping Xin[1], Xianjun Jiang[1]

[1]College of Resources and Environment, Southwest University, 2 Tiansheng Road, Beibei, Chongqing 400715, China
[2]Biogeochemistry and Nutrient Cycling Laboratory, Department of Land, Air and Water Resources, University of California Davis, CA 95616, USA

*Correspondence to*: Xianjun Jiang (jiangxj@swu.edu.cn)

**Abstract.** Iron (Fe) affects soil nitrogen (N) cycling processes both in anoxic and oxic environments. The role of Fe on soil N transformations including nitrification, mineralization, and immobilization, is influenced by redox activity, which is regulated by soil pH. The role of Fe minerals, particularly oxides, on affecting soil N transformation processes depends on soil pH, with Fe oxide often stimulating nitrification activity in the soil with low pH. We conducted lab incubations to investigate the effect of Fe oxide on N transformation rates in two subtropical agricultural soils with low pH (pH 5.1) and high pH (pH 7.8). $^{15}$N-labelled ammonium and nitrate were used separately to determine N transformation rates combined with Fe oxide (ferrihydrite) addition. Iron oxide stimulated net nitrification in low pH soil (pH 5.1), while the opposite occurred in high pH soil (pH 7.8). Compared to the control, Fe oxide decreased microbial immobilization of inorganic N by 50% in low pH soil but increased it by 45% in high pH soil. A likely explanation for the effects at low pH is that Fe oxide increased $NH_3$-N availability by stimulating N mineralization and inhibiting N immobilization. These results indicate that Fe oxide plays an important role in soil N transformation processes and the magnitude of the effect of Fe oxide is depending significantly on soil pH.

**Keywords:** Fe redox, net nitrification, gross mineralization, microbial immobilization

## 1 Introduction

The effect of soil pH on redox sensitive of soil N transformations, especially nitrification, is receiving increasing attention. Nitrification is a biological process that spans the full range of oxidation states of nitrogen (N) from –3 (ammonium: $NH_4^+$) to +5 (nitrate: $NO_3^-$), including compounds with intermediary oxidation states such as hydroxylamine ($NH_2OH$) and nitrite ($NO_2^-$) which are formed to various degrees during nitrification. This process also can produce reactive intermediates such as nitrogen oxides (NOx) and nitrous oxide ($N_2O$) affecting air quality and global climate. The role of iron (Fe) oxides on soil nitrification process both in anoxic (Clément et al., 2005; Yang et al., 2012; Ding et al., 2014) and oxic environments (Jiang et al., 2015a) is recognized, yet is rarely identified in biogeochemical models that predict global N cycle processes. Iron and

its oxides are found in abundance in many soils, with large amounts of Fe oxides are typically found in subtropical and tropical soils. Thus, understanding of the relationship between Fe oxides and soil nitrification process is especially important for understanding its influence on N cycling processes.

The direct participation of Fe in nitrification was first proposed as the Feammox reaction (Li et al., 1988; Sawayama, 2006), referred to as anaerobic $NH_4^+$ oxidation coupled to Fe(III) reduction. Nitrate and dinitrogen ($N_2$) are produced during this process (Luther et al., 1997; Clément et al., 2005; Sawayama, 2006; Shrestha et al., 2009; Yang et al., 2012). Feammox usually occurs in anoxic conditions of saturated soils such as wetland (Clément et al., 2005; Shrestha et al., 2009), suggesting that Fe oxides can act as an electron acceptor and play a critical role influencing N reactions in the absence of oxygen ($O_2$) (Schuur and Matson, 2001; Wang and Newman, 2008; Liptzin and Silver, 2009; Park et al., 2009; Ding et al., 2014).

Iron can also participate in other soil N transformations (e.g. N mineralization, heterotrophic denitrification, and chemodenitrification) via the Fe reduction-oxidation (redox) cycle, both biotically and abiotically (Li et al., 2012; Zhu-Barker et al., 2015). In ten agricultural soils, Zhu et al. (2013) found that Fe ranked higher than any other intrinsic soil property in affecting nitrous oxide ($N_2O$) production and emission. In wetland soils, sediments or anoxic microsites, oxidation of Fe(II) coupled to $NO_3^-$ reduction proceed simultaneously via biotic and abiotic pathways (Senn and Hemond, 2002; Davidson et al., 2003; Straub et al., 2004; Weber et al., 2006; Smolders et al., 2010). Under oxic or anoxic conditions, the interaction between biotic (e.g. Fe(III)-reducing microorganisms) and abiotic factors (e.g. pH) mediates the redox cycle of Fe, which can lead to organic matter decomposition and thus N mineralization (Lovley and Phillips, 1986; Roden et al., 2004; Sahrawat, 2004; Weber et al., 2006; Bauer and Kappler, 2009; Hall and Silver, 2013). The oxidation of Fe(II) stimulates organic matter decomposition and is assumed to occur via two mechanisms: (1) organic matter oxidation (driven by reactive oxygen species) and acidification; (2) the release of dissolved organic carbon that can complex with Fe. A study on diverse West African rice soils showed that $NH_4^+$ production in submerged soils was significantly correlated to reducible Fe(III). It suggests that Fe-organic matter complexes are important in influencing $NH_4^+$ production in submerged soils (Sahrawat, 2004).

Iron oxides can affect microbial groups and their activities. Meiklejohn (1953) found that a small amount of Fe (0.1-6 mg $L^{-1}$) stimulated growth of nitrifying bacteria and increased the oxidation of $NH_3$ to $NO_2^-$, whereas high concentrations of Fe (>112 mg $L^{-1}$) were toxic to nitrifying bacteria. Studies on Fe requirements for ammonia-oxidizing bacterial (AOB) showed that when the Fe concentration in the medium of *Nitrosomonas europaea* culture increased from 0.2 to 10 μM Fe, the activities of both ammonia monooxygenase and hydroxylamine oxidoreductase decreased (Wei et al., 2006). A recent study observed that the abundance of AOB and ammonia-oxidizing archaeal (AOA) in an acidic forest soil decreased after the addition of hematite, a type of Fe oxide (Jiang et al., 2015a). Nevertheless, it is difficult to generalize the response of nitrification to Fe oxide addition under varying soil pH. This is because AOA and AOB occupy different soil niches according to soil pH, i.e., AOA dominates nitrification activity in acidic soils while AOB in alkaline soils (Stopnišek et al., 2010; Gubry-Rangin et al., 2011; Isobe et al., 2012; Jiang et al., 2015b).

We hypothesized that the effect of Fe oxide on N transformations depends on a large extent on soil pH. Two major questions are posed: i) Does the presence of Fe oxide influence the rate and amount of nitrification, N mineralization, and N immobilization in soils with varying pH? ii) What is the mechanism of Fe oxide that influenced N transformations under different soil pH? To investigate Fe oxide affects on N dynamics in soils with different pH, a stable isotope ($^{15}$N) method was used to measure the gross rates of N transformations.

## 2 Material and Methods

### 2.1 Site description and soil sampling

Field sites are located at Beibei, Chongqing, China, which have a mean annual temperature of 18.2 ℃, annual rainfall of 116 cm, and a frost-free period of 359 days per year. Soil samples were collected from agricultural land (29.70 °N, 106.38 °E) with low pH soil (pH 5.1) and a hill site (29.75 °N, 106.40 °E) with high pH soil (pH 7.8) in March, 2015. Both soils were developed from a Cenozoic Quaternary Holocene (Q4) alluvium and are classified as Fluvents, Udifluvents (USDA, soil taxonomy) (Soil Survey Staff, 2014). The low pH soil was sampled from maize plots in a rotation system with sweet potato under conventional cultivation over ten years. In spring maize and autumn sweet potato growing seasons, N fertilizers were conventionally applied as urea at rates of 75 and 225 kg N ha$^{-1}$, respectively. The high pH soil was sampled from a pear orchard, which was converted from cropland three years ago and never been fertilized or tilled since the conversion. Composite soil samples derived from five auger borings to 0-20 cm in depth were brought immediately to the laboratory. Stones, dead plant material, roots, and visible soil fauna were removed. One portion of the soil was slightly air-dried to reach a gravimentric moisture content of about 15 %, sieved to 2 mm, and stored at 4 ℃ prior to use (within two months). Another portion of the soil was air-dried, passed through a 1 mm sieve and used for chemical analyses (Jiang et al., 2015a).

### 2.2 Soil chemical analyses

The results for the soil chemical properties are shown in Table 1. Soil pH was measured using a soil to water ratio of 1:2.5 (v/v) by a DMP-2 mV/pH detector (Quark Ltd, Nanjing, China). Total N (TN) and soil organic matter (SOM) contents were determined by a Macro Elemental Analyzer (Elementar Analysensysteme GmbH, Hanau, Germany). Total soil Fe was extracted with $HNO_3$-$HF$-$HClO_4$ and measured by atomic absorption spectrophotometry with a graphite furnace (GFAAS) using a model Z-8200 spectrophotometer. Available Fe was extracted using the diethylenetriamine penta-acetic acid (DTPA) method and analyzed by Inductively Coupled Plasma Optical Emission Spectrometry (ICP-OES).

### 2.3 Preparation of Fe oxide treatments

Ferrihydrite was used as a precursor to produce Fe oxide. The method for ferrihydrite preparation in this study was modified from the method described by Lovley and Phillips (1986). Briefly, Fe(III) sulfate hydrate ($Fe_2S_3O_{12}$ x$H_2O$) (40 g) and ultrapure water (500 ml) were first mixed and stirred. Then, the pH of mixture was adjusted to 7–8 with 1 mol L$^{-1}$ KOH and

left to settle until entirely precipitated. The precipitate was centrifuged (2800 g, 5 min) and washed with ultrapure water five times until the suspension had a conductivity of < 20 μS. The particle density and original pH of ferrihydrite in the final suspension were 87 g $L^{-1}$ and 3.7, respectively. One portion of the suspension was freeze-dried and then analyzed for ferrihydrite using the X-ray diffractometer (XRD) (PANalytical B.V., Holland) (Fig. 1). The remaining ferrihydrite suspension was divided into two parts. The pH of these two parts were adjusted to either 5.1 or 7.8, the same as the initial pH in low pH or high pH soils.

For each soil, two Fe oxide treatments were applied: non-Fe (control) and Fe treated (the pH of ferrihydrite was adjusted to the same as soil pH). Ferrihydrite was added at 3 % (w/w). The low pH soil without or with Fe oxide amendment was designated as pH 5.1 control or pH 5.1+Fe, while the high pH soil without or with Fe oxide amendment was designated as pH 7.8 control or pH 7.8+Fe. The suspension of ferrihydrite was added according to the treatment design and mixed well with soils. The soil mixtures were then slightly air-dried to reach a gravimentric moisture content of about 15 %, and passed through a sieve of 2 mm and stored at 4 ℃ before use (within 7 days).

### 2.4 Soil incubation with $^{15}$N substrates

In this study, a set of $^{15}$N tracing experiments were conducted to quantify process-specific and pool-specific N transformation rates. For each soil, Fe treated or non-Fe treated soils were weighed (20 g dry mass) into 150-ml conical flasks. Two N treatments were applied, $^{15}$N enriched $(NH_4)_2SO_4$ (10 atom% excess) or $^{15}$N enriched $KNO_3$ (10 atom% excess). Each N treatment received 50 mg N $kg^{-1}$ soil at the beginning of incubation. All treatments were incubated at 100 % water holding capacity (WHC) at 28 ℃ for 6 days in dark after N application. The experimental design and treatment application was set as a completely randomized block design, with three replicates per treatment (120 total experimental units comprising 5 soil sampling times). 100% WHC was chosen to create an oxic-anoxic interface, in which the redox cycle of Fe oxide commonly exists. All the flasks were covered with polyethylene film punctured with needle holes to maintain oxic conditions in the headspace.

### 2.5 Soil extraction and soil N analysis

For soil mineral N analysis, three replicates for each treatment were extracted with 2 mol $L^{-1}$ KCl (5 to 1 extractant volume to soil mass ratio) at hour 0 and 0.5, and days 1, 3, and 6 after N application. The extracted soils were centrifuged at 1200 rpm and the supernatants were frozen at –20 ℃ until analysis. The contents of $NH_4^+$ and $NO_3^-$ were quantified colorimetrically on a GENESYS 10 UV spectrophotometer (ThermoScientific, Madison, WI) using the salicylate method and the single reagent method, respectively (Verdouw et al., 1978; Doane and Horwath, 2003). Isotope analysis of $NH_4^+$ and $NO_3^-$ was performed on aliquots of the extracts using a diffusion technique, by which $NH_4^+$ was distilled with Mg oxide and $NO_3^-$ was converted to $NH_4^+$ by Devarda's alloy and then distilled with Mg oxide. The $NH_3$ volatilizes were trapped using a boric acid solution (Feast and Dennis, 1996; Zhang et al., 2011, 2013). The $^{15}$N isotopic composition in the trapped $NH_3$

volatilizes were then analyzed using an automated C/N analyzer coupled to an isotope ratio mass spectrometer (Europa Scientific Integra, UK).

Soils were fumigated or unfumigated then extracted with 0.5 mol $L^{-1}$ $K_2SO_4$ (5 to 1 extractant volume to soil mass ratio) at time 0 and 6 days after N application for soil microbial biomass N (MBN) determination (Brookes et al., 1985; Breland and Hansen, 1996; Dempster et al., 2012). The extracts were filtered, and the supernatant stored at –20 ℃ until analysis. The total dissolved N (TDN) in the extractant was separated by distillation with 25 mol $L^{-1}$ NaOH solution (Brooks et al., 1989). The $^{15}$N isotopic composition in TDN was analyzed using an automated C/N analyzer coupled to an isotope ratio mass spectrometer (Europa Scientific Integra, UK). MBN (calculated from 1 day CHCl$_3$–N) was calculated as the difference of TDN between fumigated and un-fumigated soils (Brookes et al., 1985; Dempster et al., 2012).

## 2.6 Analysis of Fe(II) production

Reduced iron, Fe(II), was quantified using the ferrozine assay method (Stookey, 1970). Briefly, 0.1 g soil was extracted with 0.5 mol $L^{-1}$ HCl (Lovley and Phillips, 1987) and 100 μl of extracts was added to 4 ml of color reagent (1 g $L^{-1}$ Ferrozine in 50 mmol $L^{-1}$ HEPES buffer pH 8). After the color developed (approximately in 15 s), the ferrous concentration was spectrophotometrically determined immediately by measuring the absorbance of the ferrozine-Fe(II) complex at 562 nm. Standards of ferrous iron for the ferrozine assay were prepared with ferrous ethylene diammonium sulfate dissolved in 0.5 mol $L^{-1}$ HCl (Lovley and Phillips, 1986).

## 2.7 Data calculation

Gross N mineralization rate was calculated according to Kirkham and Bartholomew (1954) and Davidson et al. (1991). Net nitrification rate was calculated from the net increase in $NO_3^-$ concentration in the $(NH_4)_2SO_4$ treatment during the incubation period (Davidson et al., 1992). Microbial biomass $^{15}$N (MB$^{15}$N) was calculated as MB$^{15}$N = F$^{15}$N/0.54 (Brookes et al., 1985), where F$^{15}$N = (TD$^{15}$N in the digested fumigated sample) – (TD$^{15}$N in the digested non-fumigated sample). Total dissolved $^{15}$N (TD$^{15}$N) of fumigated and un-fumigated soils were calculated by multiplying the atom% excess TD$^{15}$N and the amount of N in the form of TDN (Shen et al., 1984; Brookes et al., 1985).

## 2.8 Statistical analyses

Differences in soil $NH_4^+$ and $NO_3^-$ content, net nitrification rate, gross mineralization rate, and MBN content among different treatments were assessed by analysis of variance (ANOVA). Prior to any statistical analysis, the normality of the data was evaluated by Shapiro–Wilk test and appropriate transformation (e.g. natural log-transformation) of the data was carried out if the transformation improved the normality. Post hoc Tukey's honestly significant difference multiple comparisons of means or paired t tests were used when appropriate to verify significant differences ($P<0.05$) between treatments. All statistical analyses were performed by SPSS statistical package.

# 3 Results

## 3.1 Soil inorganic N concentrations during the incubation

The dynamics of soil inorganic N concentrations during the 6-day incubation are shown in Fig. 2. In both low and high pH soils with $(NH_4)_2SO_4$ application, the $NH_4^+$-N concentrations significantly decreased over the course of incubation. For example, in the high pH soil with Fe oxide and $(NH_4)_2SO_4$ were applied, the $NH_4^+$-N concentrations were 30.9 and 15.6 mg N $kg^{-1}$ soil at day 1 and 6, respectively (F=39.1, P=0.003). The $NO_3^-$-N concentrations increased significantly in all the $(NH_4)_2SO_4$ treatments during the incubation. However, the $NO_3^-$-N concentrations in all the $KNO_3$ treatments did not fluctuated significantly during the course of incubation ($P > 0.05$) (Fig. 2c and 2d).

## 3.2 Gross N mineralization and net nitrification rates

The gross N mineralization rate in the high pH soil was significant higher in the control than in the Fe treated soil, whereas the opposite occurred in the low pH soil (Fig. 3a) ($P < 0.05$). During the entire incubation, 22.4 and 7.80 mg $NH_4^+$-N $kg^{-1}$ was mineralized in the high pH soil without and with Fe oxide amendment, while 5.88 and 7.32 mg $NH_4^+$-N $kg^{-1}$ was mineralized in the low pH soil without and with Fe oxide amendment, respectively. No difference in gross N mineralization rate was found between the low and high pH soils with Fe oxide amendment, but the non-Fe treated (control) high pH soil had the highest gross N mineralization among all the treatments.

The net nitrification rate in the high pH soil without Fe oxide amendment was 6.02 mg N $kg^{-1}$ soil $day^{-1}$, the highest among all the treatments, while the smallest net nitrification rate was 2.41 mg N $kg^{-1}$ soil $day^{-1}$ in the low pH soil without Fe oxide amendment. Compared with the control, the addition of Fe oxide significantly decreased the net nitrification rate in the high pH soil by 22.7%, whereas 27.1% of net nitrification rate was increased by Fe oxide in the low pH soil (F = 63.1; P = 0.048) (Fig. 3b).

## 3.3 Microbial nitrogen immobilization

The $(^{15}NH_4)_2SO_4$ amended soils were used to determine microbial N immobilization as affected by Fe oxide in the 6-day incubation (Fig. 4). The $^{15}N$ content in $MB^{15}N$ was 0.17 mg N $kg^{-1}$ soil in the low pH soil with Fe oxide amendment, which was significantly lower than in the high pH soil with Fe oxide amendment (0.65 mg N $kg^{-1}$ soil). The addition of Fe oxide had no significant influence on $MB^{15}N$ in the low pH soil, while $MB^{15}N$ content in the high pH soil was 3.7 times higher in the Fe oxide treatment than in the control (Fig. 4a). In the low pH soil, the total N in MBN was 13.8 mg N $kg^{-1}$ soil in the control, which was significant lower than it in the high pH soil without Fe oxide amendment (15.2 mg N $kg^{-1}$ soil). The addition of Fe oxide caused a significant decrease in MBN in the low pH soil, while the opposite occurred in the high pH soil ($P < 0.05$) (Fig. 4b). Compared with control, Fe oxide addition decreased MBN by 50% in the low pH soil but increased it by 45% in the high pH soil.

## 3.4 Fe(II) production

The concentration of Fe(II) (0.5 mol L$^{-1}$ HCl extractable) in the soils with Fe oxide amendment before and after 6 days incubation are shown in Fig. 5. In the low pH soil amended with Fe oxide, the concentration of Fe(II) was increased from 0.44 to 1.28 mg Fe kg$^{-1}$ soil after the 6 days incubation. In the high pH soil amended with Fe oxide, the concentration of Fe(II) did not change between day 0 and days 6.

## 4 Discussion

In the present study, the addition of Fe oxide increased the net nitrification rate by 20.8 % in the low pH soil. The effect of Fe oxide on the nitrification rate varied with soil pH, supporting our hypothesis. Nitrification is primarily dependent on NH$_3$ availability and the activity of nitrifying microorganisms. A regression-analysis assessing relationships among the rates of N transformation processes from 100 published studies with nearly 300 different organic and mineral soil materials concluded that nitrification rate is controlled by the rate of ammonia released from soil organic matter mineralization (Booth et al. 2005). The addition of Fe oxide had opposite effects on nitrification, stimulating it at low pH soil (pH 5.1) (F = 63.13; P = 0.048) and lowering it at high pH soil (pH 7.8).

The amount of substrate ammonia available for nitrification is dependent on the gross N mineralization rate. Gross N mineralization increased significantly in the low pH soil with Fe oxide. An increase in N mineralization (R–NH$_2$ → NH$_3$) likely increased the availability of NH$_3$, leading to an increase in nitrification. Generally, both AOA and AOB play roles in nitrification, but it is difficult for AOB to sustain ammonia oxidation in soil with low pH due to the high pKa of ammonia (NH$_3$ + H$^+$ → NH$_4^+$; pKa = 9.25) (Kuroiwa et al., 2011). Since AOA have much higher affinity for NH$_3$ than AOB (Martens-Habbena et al., 2009), it generally dominates nitrification activity in acidic soils (Stopnišek et al., 2010; Gubry-Rangin et al., 2011; Isobe et al., 2012; Prosser and Nicol, 2012; Jiang et al., 2015b).

The high solubility of Fe(III) at low pH could also promote scavenging of hydroxylamine (NH$_2$OH), an intermediate in nitrification (Vajrala et al., 2013), by the chemical reaction 2Fe$^{3+}$ + 2NH$_2$OH → 2Fe$^{2+}$ + N$_2$ + 2H$_2$O + 4H$^+$ (Zhu-Barker et al., 2015). This assumption was supported by the significant increase in Fe(II) concentration at the end of the incubation in the low pH soil with Fe oxide amendment (Fig. 5). Under low pH, Fe(III) solubility is generally high (Weber et al., 2006). It was not absolutely certain that the 100 % WHC soil moisture in our incubation provided a complete anoxic environment for the occurrence of anaerobic NH$_4^+$ oxidation into NO$_3^-$ or NO$_2^-$ by coupled Fe(III) reduction (Feammox). Thus, the process of Feammox cannot be used to explain exclusively the increase in the net nitrification rate in low pH soil. Further studies are needed to fully understand the process of Feammox in the low pH soil.

In the high pH soil, iron oxide significantly decreased the net nitrification rate, likely due to increased inorganic N immobilization (Fig. 4b). Furthermore, the toxicity of Fe oxide on nitrifying microorganisms can be another important reason for Fe oxide decreasing nitrification at high pH. For example, Meiklejohn (1953) demonstrated that high Fe (> 112 mg L$^{-1}$) was toxic to nitrifying bacteria.

Previous research showed that soil microbial communities prefer $NH_4^+$ to $NO_3^-$ as a source of N (Jansson et al., 1955; Recous et al., 1990; Zhang et al., 2013). However, $NO_3^-$ immobilization was found to be high in undisturbed forest soils (Stark and Hart, 1997), suggesting that microbial biomass maybe flexible in utilizing different N sources. Besides the factors of inorganic N ($NH_4^+$ or $NO_3^-$) availability and microbial activity (Stark and Hart, 1997), Fe oxide is another factor affecting microbial N immobilization. In the low pH soil, the addition of Fe oxide caused a significant decrease in MBN, while the opposite was found in the high pH soil (Fig. 4b). This indicates that high solubility of Fe oxide at low pH environment can impair the assimilation of N and reduce the size of the microbial biomass.

While Fe oxide in the low pH soil decreased microbial N assimilation, the Fe(III) reduction process can release Fe-bound N and lead to an increase in N mineralization and ammonification, thus increase nitrification potential. The addition of Fe oxide had no influence on $MB^{15}N$ in the low pH soil, whereas in the high pH soil, 3.7 times higher $MB^{15}N$ was found in the Fe oxide treatment than in the control (Fig. 4a). The high $MB^{15}N$ content in the high pH soil with Fe oxide addition was probably related to the low activity of Fe oxide in the high pH soil due to the low solubility of Fe(III) oxide (Weber et al., 2006). Further research is needed to explore the mechanism of how the addition of Fe oxide increases microbial N assimilation in the high pH environment.

**5 Conclusions**

The addition of Fe oxide stimulated net nitrification and gross N mineralization rates but reduced microbial N immobilization in low pH soil. The opposite was observed in high pH soil. These findings indicated that Fe oxide has an important role in soil N transformations. The effect of Fe oxide on N transformations varies with pH. Further studies should focus on Fe redox in different pH soils to develop the mechanistic understanding of how Fe oxide changes N mineralization and nitrification through abiotic and biotic-related processes to influence the production of $N_2$, $N_2O$ and $NO_2^-$.

**Author contributions**

Xueru Huang and Xianjun Jiang designed the experiments and Xueru Huang carried them out. Xueru Huang prepared the manuscript with contributions from co-authors. Xia Zhu-Barker revised and edited the manuscript. Xianjun Jiang, Xia Zhu-Barker, William R. Horwath and Sarwee J. Faeflen helped with writing and English language checking.

**Acknowledgements**

This work was supported by the National Natural Science Foundation of China (41271267).

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

**Table 1.** Chemical properties of the soils with low pH and high pH. Different letters represent statistically significant between treatments at $P < 0.05$.

| Soil type | pH | Organic matter g kg$^{-1}$ | Total N g kg$^{-1}$ | Total Fe g kg$^{-1}$ | Available Fe mg kg$^{-1}$ | NO$_3^-$-N mg N kg$^{-1}$ | NH$_4^+$-N mg N kg$^{-1}$ |
|---|---|---|---|---|---|---|---|
| Fluvents Udifluvents | 5.1 | 18.0 ±0.26 a | 0.73 ±0.01 a | 16.3 ±0.08 b | 132 ±4.04 a | 10.3 ±0.85 a | 1.54 ±0.19 b |
| Fluvents Udifluvents | 7.8 | 13.9 ±0.11 b | 0.68 ±0.03 a | 27.5 ±0.04 a | 5.64 ±0.49 b | 4.68 ±0.48 b | 2.44 ±0.16 a |

**Figure Captions**

**Figure 1:** X-ray diffraction pattern of ferrihydrite.

**Figure 2:** Effects of Fe oxide on $NH_4^+$-N and $NO_3^-$-N dynamics during 6-day by $^{15}N$ tracing incubation at 28 ℃ with soil moisture at 100 % WHC. $NH_4^+$-N and $NO_3^-$-N concentrations were measured following the addition of 50 mg N $kg^{-1}$ $(^{15}NH_4)_2SO_4$ (a and b) and $K^{15}NO_3$ (c and d). Error bars represent standard deviation, n = 3.

**Figure 3:** Effects of Fe oxide on gross mineralization rate (a) and net nitrification rate (b) during 6-day for incubated soil samples incubation at 28 ℃ with soil moisture at 100 % WHC. Error bars represent standard deviations, n = 3. The different letters above the columns indicate a significant difference ($P < 0.05$).

**Figure 4:** Effects of Fe oxide on $MB^{15}N$ (a) and MBN (b) pools during 6-day with $(^{15}NH_4)_2SO_4$ treatment incubation at 28 ℃ with soil moisture at 100 % WHC. Error bars represent standard deviations, n = 3. The different letters above the columns indicate a significant difference ($P < 0.05$).

**Figure 5:** Effects of Fe oxide on concentration of Fe(II) (0.5 mol $L^{-1}$ HCl extractable) before and after 6-day with $(^{15}NH_4)_2SO_4$ treatment incubation at 28 ℃ with soil moisture at 100 % WHC. Error bars represent standard deviations, n = 3. The different letters above the columns indicate a significant difference ($P < 0.05$).

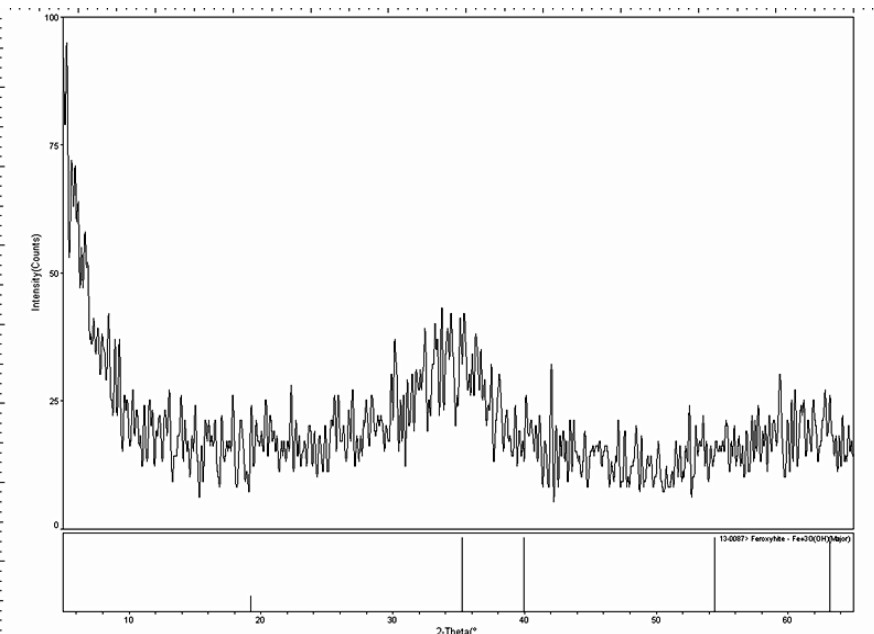

**Figure 1:** X-ray diffraction pattern of Ferrihydrite.

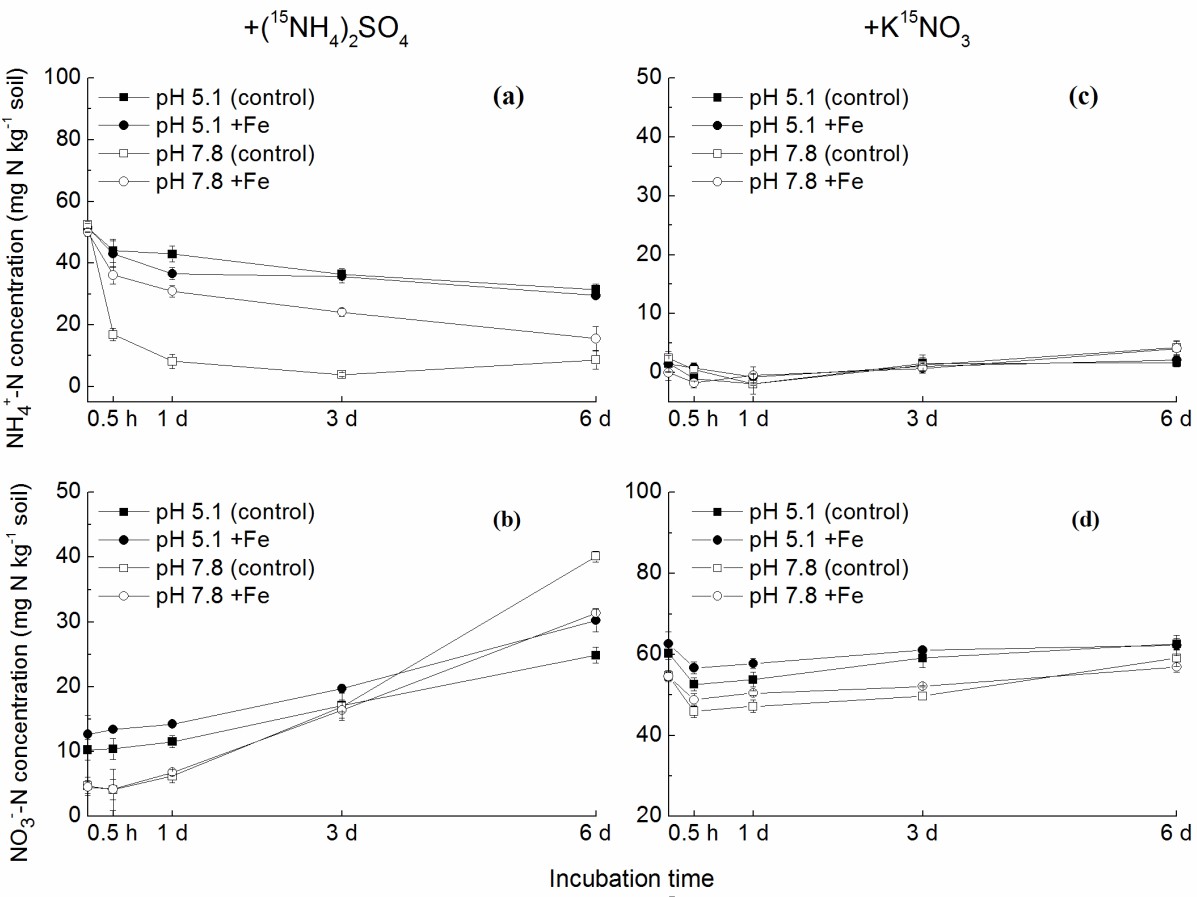

**Figure 2:** Effects of Fe oxide on $NH_4^+$-N and $NO_3^-$-N dynamics during 6-day by $^{15}$N tracing incubation at 28 ℃ with soil moisture at 100 % WHC. $NH_4^+$-N and $NO_3^-$-N concentrations were measured following the addition of 50 mg N kg$^{-1}$ ($^{15}NH_4$)$_2SO_4$ (a and b) and K$^{15}NO_3$ (c and d). Error bars represent standard deviation, n = 3.

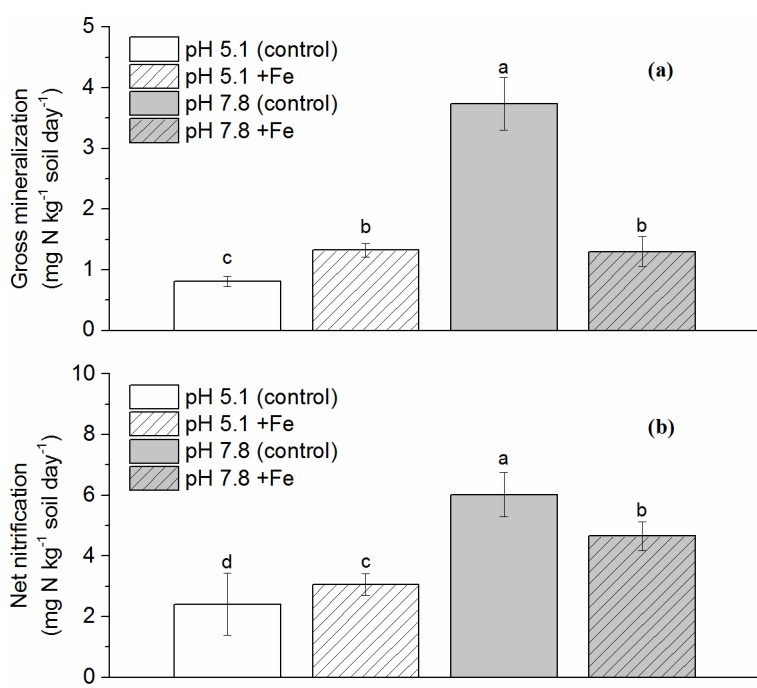

**Figure 3:** Effects of Fe oxide on gross mineralization rate (a) and net nitrification rate (b) during 6-day for incubated soil samples incubation at 28 ℃ with soil moisture at 100 % WHC. Error bars represent standard deviations, n = 3. The different letters above the columns indicate a significant difference ($P < 0.05$).

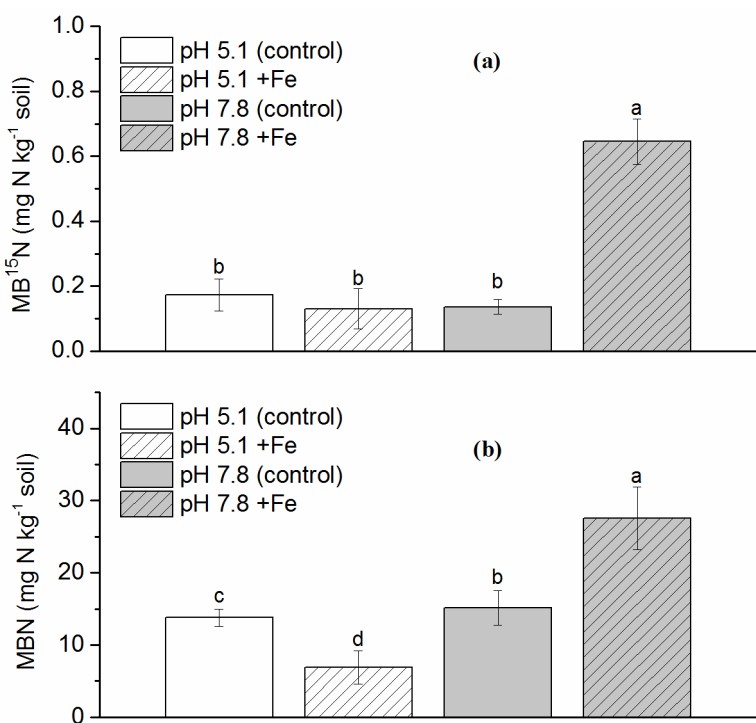

**Figure 4:** Effects of Fe oxide on MB$^{15}$N (a) and MBN (b) pools during 6-day with ($^{15}$NH$_4$)$_2$SO$_4$ treatment incubation at 28 ℃ with soil moisture at 100 % WHC. Error bars represent standard deviations, n = 3. The different letters above the columns indicate a significant difference ($P < 0.05$).

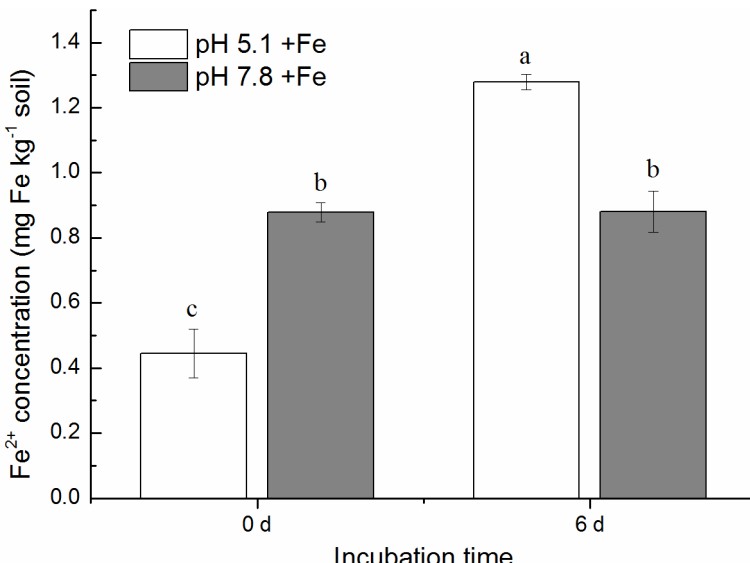

**Figure 5:** Effects of Fe oxide on concentration of Fe(II) (0.5 mol $L^{-1}$ HCl extractable) before and after 6-day with
($^{15}NH_4$)$_2SO_4$ treatment incubation at 28 ℃ with soil moisture at 100 % WHC. Error bars represent standard deviations, n = 3. The different letters above the columns indicate a significant difference ($P < 0.05$).
