# Peer review of "Effect of iron oxide on nitrification in two agricultural soils with different pH"

_Biogeosciences, 2016_

## Referee Comment (RC1) · Anonymous Referee #1 · 18 Jul 2016

The manuscript focuses on effects of iron oxide on nitrification in two agricultural soils with different pH, which is within the scope of Biogeoscience. Nitrification is a key process in the global nitrogen cycle. This paper has an interesting topic and using 15N stable isotope method in this study is appropriate for assessing iron oxide effects on net nitrification, gross mineralization, and microbial immobilization. However, parts of the manuscript are unclear, missing key information, and require further clarifications and better interpretation. This manuscript would also benefit from language editing by a native speaker. Specific comments:

1. P. 1, L. 9-19: The abstract needs to be more descriptive. Variations in what way? 2. P. 3, L. 64: Please show the date of soil sampling and management history of the land. 3. P3., L70 and Table 1. Statistical data is missing from Table 1. What is "Available Fe" in Table 1? How was it determined? In addition, redox potential (Eh)

[Figure]

of soil is important to understanding your data, but it is missing. 4. P. 3, L. 77-79: You adjusted the pH of ferrihydrite suspensions to 5.1 and 7.8, respectively, by using KOH. What's the original pH of ferrihydrite suspension? It would be helpful if some basic properties of the Fe oxide were measured, such as specific surface area, zero point of charge, cation exchange capacity and anion exchange capacity. Moreover, X-ray diffraction analysis was performed, but this information was not presented and discussed in the results and discussion sections. 5. P. 3, L. 89-91: More details are needed regarding the measurements of total Fe and free Fe oxides. Free Fe oxide data was not presented in the results and discussion sections. Please note "free Fe oxide" in soil cannot be used to represent "available Fe". 6. P. 4, L. 92-98: It is not clear as to what experimental design was used in this study. 7. P. 4, L. 97: Please justify why soil moisture content was adjusted to 100% WHC? 8. P. 5, L. 133-136 and Figure 1: LSD test is needed in Figure 1, especially if you want to show a significant decrease in NH4-N. 9. General comments on the Results and Discussion sections: In the acidic soil, amendment of Fe oxide resulted in a decrease in microbial biomass, likely due to accumulation of $Fe^{2+}$ (Figure 4). 10. P6 L. 164-165: The addition of Fe oxide stimulated the net nitrification rate in the low pH soil (pH 5.1) (F = 63.13; P = 0.048), but suppressed it in the high pH soil (pH 7.8). In the acidic soil, amendment of Fe oxide resulted in a decrease in microbial biomass (Fig. 3), due to toxic effect of $Fe^{2+}$(Fig. 4). However, the increased gross mineralization and nitrification in the Fe oxide amended soil (Fig. 2) seems to conflict with the decreased microbial biomass (Fig. 4). Similarly, in the high pH soil (7.8), it is difficult to understand that enhanced microbial biomass in the Fe oxide amended soil (Fig. 4) would result in decreased gross mineralization and nitrification.

In general, at pH7.8, Fe oxide in soil is quite inertial. The significant decrease in gross mineralization and nitrification and a significant increase in microbial biomass by amendment of 3% Fe oxide are unexpected.

---

## Referee Comment (RC2) · Anonymous Referee #2 · 20 Jul 2016

This article examines how pH and iron oxides interact while impacting the effective processes of nitrification, mineralization and immobilization in subtropical agricultural soils under anoxic conditions. The science is good, the article is short and concise (which is good), and is typically in the scope of BGS. In particular, targeting how iron oxides impact nitrification, a key process in many soils but especially in tropical soils where the presence of iron is important, is crucial and little addressed in the literature.

Overall, the English is to be improved (even if not catastrophic); ask a native English speaker or equivalent to proofread the manuscript. Finally, a number of points are also to be improved, listed below:

ABSTRACT: * Line 13: here, and later in the manuscript: please specify the reason why the experiment was done at 100% WHC; * In the abstract in general: Avoid vague

[Figure]

phrases like eg. 'We hypothesized that the effect of Fe oxide on N transformation processes would be different' (line 11); be more specific about the expected effects;

INTRODUCTION: * Here, and in the discussion, there are no details on the potential impact of the processes studied on denitrification rates. Specify in a few lines how your experimental conditions, or the presence of iron in general, likely impact denitrification under anoxic conditions; * Line 23: 'affect'->'affects'; * Lines 23-25: yes, I do agree that the role of iron on nitrification is important and little studied. Also specify how this is especially important for tropical or subtropical ecosystems; * Line 38: 'such as humic substances'-> please specify if this parameter's control on nitrification is through quantity, quality, both?; * Line 40: 'Meiklejohn, 1953'-> please find a more recent reference; * Lines 42-43: 'These findings confirm the relevance of Fe oxides as a key factor in promoting pathways leading to N loss in soils.'-> this is not clear to me. You state in the sentence before that hematite is lowering AOB and AOA, so it is likely not promoting but lowering N loss, as denitrification should be reduced due to lower nitrification…; * Lines 47-49: I don't understand the difference between the two questions: 'Does the presence of Fe oxide influence the rate and amount of nitrification, N mineralization, and N immobilization in soils with different pH?'; and 'How does Fe oxide influence these N transformation processes under different pH in soils with 100 % water holding capacity (WHC)?'. Please be more specific.

MATERIAL AND METHODS: * Line 54: 'days->'days per year'; * Lines 53-59: please be more precise with the description of the studied sites. - Soils are classified as Fluvents, Udifluvents: both (agricultural land vs. hill)? Please describe what it means; - Precise soils management (possibly by citing previous papers on these sites); especially for the high pH site: how many years ('a few') after conversion? What was the amount of N fertilizer before? - Precise the dates of sampling; * Line 62: 'and stored at 4 °C prior to use'-> for how long exactly?; * Line 63: 'passed through a 1 mm sieve'->why not 2mm? * Lines 63-64: 'The results of the chemical properties of soils are shown in Table 1.'-> please put this sentence at the beginning of the 'Soil chemical

parameter' paragraph; * Line 71: 'XRD': please specify the brand of device; * Line 73 and below: please choose a constant term in the manuscript between 'amended with Fe' vs. +Fe' vs. 'Fe treated'; * Line 77: why sieving at 2mm again? (the second time); * Line 82:' free Fe oxides'-> what are they, 'available Fe' in Table 1? If not, they are lacking in Table 1 and 'available Fe' are not described; * Please put paragraph '2.3 Soil chemical analyses' before paragraph '2.2 Preparation of Fe oxide treatments'; * Paragraph '2.4 Experimental design and 15N addition': please specify the total N of samples. Is it 2x2x2x3=24x5 (time kinetics)=120? * Line 90: 'incubated for 6 days at 28 °C.'-> in the dark? * Lines 96-98: please specify in few words the techniques used for colorimetry and diffusion, and the model of machines; * Line 100: 'MBN'. As for the other acronyms, once you defined them, use them always in the subsequent text; * Line 104: 'were'->'was'; * Line 110: 'aprroximately'->' approximately'; * Paragraph '2.8 Statistical analyses': please state how you have checked the normality and homoscedasticity prior to ANOVA; please also specify the post-hoc tests you used;

RESULTS: * Lines 126-128: 'In both low and high pH soils, the NH4+-N concentrations showed a significant decrease after the application of (NH4)2SO4, at both 30.9 and 15.6 mg NH4+-N kg−1 soil at day 1 and 6, respectively, in the higher pH soil with the Fe oxide amendment' -> not clear, please rephrase; * Lines 126-128: please describe the +K15NO3 figures and results; * Lines 137-140: please state what is significant. . .; * Line 145: 'but slightly decreased it in the low pH soil'-> no, it is not significant so there is no decrease;

DISCUSSION: * Line 158: 'suppressed'-> too strong. 'lowered'?; * Line 168: 'Kuroiwa et al., 2011'-> reference lacking; * Line 169: 'it dominates nitrification'->' it generally dominates nitrification'; * Line 176: 'occurance'->' occurrence'; * Generally in the Discussion: please discuss the MBN15N results. . .; and discuss the impact on denitrification process; * Line 183: 'Jansson et al., 1955'-> please find a more recent reference;

TABLE 1: * Legend: 'studied soils' too vague, precise; precise what are fluvent/udifluvents subsamples. . . the two sites? * The statistical data are lacking here! *

Specify as said above what is 'available Fe';

FIGURE 1: * Legend, line 357: 'moisture of'->'moisture at'; 'concentration'->'concentrations';

FIGURE 2: * Legend: specify Fig 2a and Fig2b after mineralisation and nitrification; * Are you sure that for Fig 2b pH5.1 control and pH 5.1 +Fe are statistically different?

FIGURE 3: * Legend: specify acronyms + Fig 3a and Fig3b after 15N and N.

---

## Author Response (AR1)

Dear Reviewers,

We thank you for your most helpful efforts in the evaluation of our manuscript.
We have uploaded a revised version of the manuscript that was extensively revised based on the reviewer comments, which we found to be very constructive and useful.
Below (Italic font) are our point-by-point responses to the reviewers' comments with references to line numbers in the revised version. Please let us know if further information or modifications are needed.
Thank you again for your expert reviewing of our manuscript.

Best wishes,

Xueru Huang (first author)
Xianjun Jiang (corresponding author)
On behalf of all the authors
* * *
Reviewer #1:

The manuscript focuses on effects of iron oxide on nitrification in two agricultural soils with different pH, which is within the scope of Biogeoscience. Nitrification is a key process in the global nitrogen cycle. This paper has an interesting topic and using 15N stable isotope method in this study is appropriate for assessing iron oxide effects on net nitrification, gross mineralization, and microbial immobilization. However, parts of the manuscript are unclear, missing key information, and require further clarifications and better interpretation. This manuscript would also benefit from language editing by a native speaker.

*We thank the reviewer#1 for your time. We have carefully considered each comment and thoroughly edited the manuscript to address each of them, with point-by-point explanations of how each comment has been addressed below. We have also substantially edited and strived to improve the English throughout the manuscript. Since considerable edits were made, we have not detailed every action here.*

**Specific comments:**

1. P. 1, L. 9-19: The abstract needs to be more descriptive. Variations in what way?

*Re: We have revised the abstract by refining the specific research purpose, describing to what extent the addition of Fe oxide increased/decreased microbial biomass N in the two soils of different pH, See page 1 lines 11-20 in the revised manuscript.*

2. P. 3, L. 64: Please show the date of soil sampling and management history of the land.

*Re: We sampled the soils on March 2015. We have added the date of* management history of the land *by "The low pH soil was sampled from maize plots in a rotation system with sweet potato under conventional cultivation over ten years. In spring maize and autumn sweet potato growing seasons, N fertilizers were conventionally applied as urea at rates of 75 and 225 kg N ha$^{-1}$, respectively. The high pH soil was*

*sampled from a pear orchard, which was converted from cropland three years ago and never been fertilized or tilled since the conversion."* on page 3 lines 75-78 of the revised manuscript.

3. P3., L70 and Table 1. Statistical data is missing from Table 1. What is "Available Fe" in Table 1? How was it determined? In addition, redox potential (Eh) of soil is important to understanding your data, but it is missing.

***Re:*** *Avaibable Fe is the Fe which can be absorbed by soil microorganisms and plants and is extracted by DPTA (e.g. Wang, C., Ji, J., Yang, Z., Chen, L., Browne, P., and Yu, R.: Effects of soil properties on the transfer of cadmium from soil to wheat in the yangtze river delta region, China-a typical industry-agriculture transition area, Bological Trace Element Research,148, 264-274, 2012). We have added the available Fe data in Table 1 and the methodology description of its measurement "Available Fe was extracted using the diethylenetriamine penta-acetic acid (DTPA) method and analyzed by Inductively Coupled Plasma Optical Emission Spectrometry (ICP-OES)."* on page 3 lines 88-89 of the revised manuscript.*

*We agree with the reviewer's comment that the redox potential (Eh) could add more information to our study and contribute to the understanding of the results. In the present study, we used the combination of soil moisture content of 100% WHC and the measurement of concentration of $Fe^{2+}$, instead of Eh, to provide robust evidence on the redox potential and understand the effect of Fe oxide on soil nitrification.*

4. P. 3, L. 77-79: You adjusted the pH of ferrihydrite suspensions to 5.1 and 7.8, respectively, by using KOH. What's the original pH of ferrihydrite suspension? It would be helpful if some basic properties of the Fe oxide were measured, such as specific surface area, zero point of charge, cation exchange capacity and anion exchange capacity. Moreover, X-ray diffraction analysis was performed, but this information was not presented and discussed in the results and discussion sections.

***Re:*** *The original pH of ferrihydrite suspension is 3.7. We revised it in "Preparation of Fe oxide treatments" section. Please see page 4 lines 95-96 in the revised manuscript. We added the figure of X-ray diffraction pattern of ferrihydrite in page 14 lines 424-425.*

*Thanks for your suggestion for the measurement of the basic properties of the Fe oxide. Since the $Fe^{2+}$ concentration can be used to understand the Fe reduction and explaining the effect of Fe oxide on soil nitrification, we did not intend to measure the basic properties of the Fe oxide, such as specific surface area, zero point of charge, cation exchange capacity and anion exchange capacity.*

5. P. 3, L. 89-91: More details are needed regarding the measurements of total Fe and free Fe oxides. Free Fe oxide data was not presented in the results and discussion sections. Please note "free Fe oxide" in soil cannot be used to represent "available Fe".

***Re:*** *"available Fe" but not "free Fe oxides" is the result of the soils chemical property in the table. We have replaced "free Fe oxides" with "available Fe" and described the analysis of available Fe in the Material and Methods section.*
*We have added the information on the measurement of available Fe "Available Fe was extracted using*

*the diethylenetriamine penta-acetic acid (DTPA) method and analyzed by Inductively Coupled Plasma Optical Emission Spectrometry (ICP-OES)." on page 3 lines 88-89 of the revised manuscript.*

6. P. 4, L. 92-98: It is not clear as to what experimental design was used in this study.

**Re:** *The experimental design and treatment application as set up as a completely randomized block design, with three replicates per treatment (120 total experimental units comprising 5 soil sampling times). Please see page 4 lines 111-113 in the revised manuscript.*

7. P. 4, L. 97: Please justify why soil moisture content was adjusted to 100% WHC?

**Re:** *100% WHC was chosen to create an oxic-anoxic interface, in which the redox cycle of Fe oxide commonly exists. We have revised justified in "2.4 Soil incubation with $^{15}N$ substrates" section. Please see page 4 lines 113-114 in the revised manuscript.*

8. P. 5, L. 133-136 and Figure 1: LSD test is needed in Figure 1, especially if you want to show a significant decrease in $NH_4$-N.

**Re:** *We agree with the reviewer that it is easier for reader to catch the statistical significant diffference among treatments and days if the statistical results are shown in the figure 1. However, there are treatment and timing two factors, we feel that showing the statistical results between each comparision treatments would complicate the figure and decrease the preference of the figure. Since we only focus on the data changes between day 1 and day 6, we have stated the statistical results in the context where is suitable. See page 6 lines 158-160 in the revised manuscript.*

9. General comments on the Results and Discussion sections: In the acidic soil, amendment of Fe oxide resulted in a decrease in microbial biomass, likely due to accumulation of $Fe^{2+}$ (Figure 4).

**Re:** *Rather than the accumulation of $Fe^{2+}$, the high solubility of Fe oxide at low pH could impair the assimilation of N by the microbial biomass, and at the meantime, the Fe(III) reduction process could release Fe-bound N and lead to N mineralization and ammonification, thus increasing nitrification potential.*

10. P. 6 L. 164-165: The addition of Fe oxide stimulated the net nitrification rate in the low pH soil (pH 5.1) (F = 63.13; P = 0.048), but suppressed it in the high pH soil (pH 7.8). In the acidic soil, amendment of Fe oxide resulted in a decrease in microbial biomass (Fig. 3), due to toxic effect of $Fe^{2+}$(Fig. 4). However, the increased gross mineralization and nitrification in the Fe oxide amended soil (Fig. 2) seems to conflict with the decreased microbial biomass (Fig. 4). Similarly, in the high pH soil (7.8), it is difficult to understand that enhanced microbial biomass in the Fe oxide amended soil (Fig. 4) would result in decreased gross mineralization and nitrification.
In general, at pH7.8, Fe oxide in soil is quite inertial. The significant decrease in gross mineralization and nitrification and a significant increase in microbial biomass by amendment of 3% Fe oxide are unexpected.

**Re:** *First, the decrease in microbial biomass N did not attribute to the toxic effect of $Fe^{2+}$, but the high*

*solubility of Fe oxide at low pH and the reduction between Fe(III) oxide and the released N. Second, the increased inorganic N from the decreased microbial immobilization of N benefits the nitrification, so it is not conflict. As we said, further studies should focus on Fe redox in different pH soils to develop the mechanistic understanding of how Fe oxide changes N mineralization and nitrification through abiotic and biotic-related processes to influence the production of $N_2$, $N_2O$ and $NO_2^-$.*
* * *
Reviewer #2:

This article examines how pH and iron oxides interact while impacting the effective processes of nitrification, mineralization and immobilization in subtropical agricultural soils under anoxic conditions. The science is good, the article is short and concise (which is good), and is typically in the scope of BGS. In particular, targeting how iron oxides impact nitrification, a key process in many soils but especially in tropical soils where the presence of iron is important, is crucial and little addressed in the literature. Overall, the English is to be improved (even if not catastrophic); ask a native English speaker or equivalent to proofread the manuscript. Finally, a number of points are also to be improved, listed below:

**Re:** *We thank the reviewer#2 for your time. We have carefully considered each comment and thoroughly edited the manuscript to address each of them, with point-by-point explanations of how each comment has been addressed below. We have also substantially edited and strived to improve the English throughout the manuscript. Since considerable edits were made, we have not detailed every action here.*

ABSTRACT:
1. Line 13: here, and later in the manuscript: please specify the reason why the experiment was done at 100% WHC;

**Re:** *100% WHC was chosen to create an oxic-anoxic interface, in which the redox cycle of Fe oxide commonly exists. We have explained it in Material and Methods Section. Please see lines 113-114 in the revised manuscript.*

2. In the abstract in general: Avoid vague phrases like eg. 'We hypothesized that the effect of Fe oxide on N transformation processes would be different' (line 11); be more specific about the expected effects;

**Re:** *We have changed this phrase to "The role of Fe minerals, particularly oxides, on affecting soil N transformation processes depends on soil pH, with Fe oxide often stimulating nitrification activity in the soil with low pH." on lines 11-12 in the revised manuscript.*

INTRODUCTION:
3. Here, and in the discussion, there are no details on the potential impact of the processes studied on denitrification rates. Specify in a few lines how your experimental conditions, or the presence of iron in general, likely impact denitrification under anoxic conditions;

**Re:** *We have revised and added the potential impact on denitrification in the Introduction section. Please see lines 27-28 and lines 42-43 in the revised manuscript.*

4. Line 23: 'affect'->'affects';

**Re:** *We have deleted the word in the revised manuscript.*

5. Lines 23-25: yes, I do agree that the role of iron on nitrification is important and little studied. Also specify how this is especially important for tropical or subtropical ecosystems;

**Re:** *We have added the sentence in Intruduction section by "Iron and its oxides are found in abundance in many soils, with large amounts of Fe oxides are typically found in subtropical and tropical soils. Thus, understanding of the relationship between Fe oxides and soil nitrification process is especially important for understanding its influence on N cycling processes." on lines 30-33 in the revised manuscript.*

6. Line 38: 'such as humic substances'-> please specify if this parameter's control on nitrification is through quantity, quality, both?;

**Re:** *In the discussion and revised edition of this manuscript, we had deleted the content "such as humic substances" in Introduction section. Even that, humic substances would control on nitrification through both quantity and quality. First, humic substances would affect soil mineralization rate so that it controls nitrification through quantity. Second, Fe reduction can be facilitated by humus substances that act as "electron shuttles", especially for dissoluble Fe oxide (e.g. ferrihydrite) reduction (Burgin, et al., 2011), so humic substances also control nitrification through quality.*

7. Line 40: 'Meiklejohn, 1953'-> please find a more recent reference;

**Re:** *We have added the information of another recent reference in Intruduction section by "Studies on Fe requirements for ammonia-oxidizing bacterial (AOB) showed that when the Fe concentration in the medium of Nitrosomonas europaea culture increased from 0.2 to 10 $\mu M$ Fe, the activities of both ammonia monooxygenase and hydroxylamine oxidoreductase decreased (Wei et al., 2006)." On lines 56-58 in the revised manuscript.*

8. Lines 42-43: 'These findings confirm the relevance of Fe oxides as a key factor in promoting pathways leading to N loss in soils.'-> this is not clear to me. You state in the sentence before that hematite is lowering AOB and AOA, so it is likely not promoting but lowering N loss, as denitrification should be reduced due to lower nitrification. . .;

**Re:** *in the discussion and revised edition of this manuscript, we had deleted the content of "These findings confirm the relevance of Fe oxides as a key factor in promoting pathways leading to N loss in soils." We had improved the paragraph in Intruduction section. Please see page 2 lines 54-63 in the revised manuscript.*

9. Lines 47-49: I don't understand the difference between the two questions: 'Does the presence of Fe oxide influence the rate and amount of nitrification, N mineralization, and N immobilization in soils with different pH?'; and 'How does Fe oxide influence these N transformation processes under different pH in soils with 100 % water holding capacity (WHC)?'. Please be more specific.

*Re: We have changed the second question by "ii) What is the mechanism of Fe oxide that influenced N transformations under different soil pH?" Please see lines 66-67 in the revised manuscript.*

MATERIAL AND METHODS:
10. Line 54: 'days->'days per year';

**Re:** *We have revised it on line 72 in the revised manuscript.*

11. Lines 53-59: please be more precise with the description of the studied sites. - Soils are classified as Fluvents, Udifluvents: both (agricultural land vs. hill)? Please describe what it means; - Precise soils management (possibly by citing previous papers on these sites); especially for the high pH site: how many years ('a few') after conversion? What was the amount of N fertilizer before? - Precise the dates of sampling;

**Re:** *Both agricultural land and hill are classified as Fluvents, Udifluvents. The low pH soil was sampled from maize plots in a rotation system with sweet potato under conventional cultivation over ten years. In spring maize and autumn sweet potato growing seasons, N fertilizers were conventionally applied as urea at rates of 75 and 225 kg N ha$^{-1}$, respectively. The high pH soil was sampled from a pear orchard, which was converted from cropland three years ago and never been fertilized or tilled since the conversion. We sampled the soils on March 2015. Please see lines 72-78 in the revised manuscript.*

12. Line 62: 'and stored at 4 °C prior to use'-> for how long exactly?;

**Re:** *Soil samples were stored at 4 °C prior to use within two months. We revised it. Please see line 81 in the revised manuscript.*

13. Line 63: 'passed through a 1 mm sieve'->why not 2 mm?

**Re:** *We have added reference (Jiang et al., 2015a) to support the method of passing soil samples through a 1 mm sieve for measuring soil chemical analyses. Please see line 82 in the revised manuscript.*

14. Lines 63-64: 'The results of the chemical properties of soils are shown in Table 1.'-> please put this sentence at the beginning of the 'Soil chemical parameter' paragraph;

**Re:** *We have revised it on line 84 in the revised manuscript.*

15. Line 71: 'XRD': please specify the brand of device;

**Re:** *We have added the brand information of device in the "2.3 Preparation of Fe oxide treatments" section by "One portion of the suspension was freeze-dried and then analyzed for ferrihydrite using the X-ray diffractometer (XRD) (PANalytical B.V., Holland) (Fig. 1)" on lines 96-97 in the revised manuscript.*

16. Line 73 and below: please choose a constant term in the manuscript between 'amended with Fe' vs. +Fe' vs. 'Fe treated';

**Re:** *We choose "Fe treated" as the constant term in the manuscript. Even that, we choose "+Fe" as the constant term in the pictures for convenience.*

17. Line 77: why sieving at 2mm again? (the second time);

**Re:** *The incubation soils were generally sieved at 2 mm, please refer to Jiang, X. J. et al. Effects of Fe oxide on N transformations in subtropical acid soils, Sci. Rep-UK, 5, 8615, 2015.*

18. Line 82:' free Fe oxides'-> what are they, 'available Fe' in Table 1? If not, they are lacking in Table 1 and 'available Fe' are not described;

**Re:** *"available Fe" but not "free Fe oxides" is the result of the soils chemical property in the table. We have replaced "free Fe oxides" with "available Fe" and described the analysis of available Fe in the Material and Methods section.*
*We have added the information on the measurement of available Fe "Available Fe was extracted using the diethylenetriamine penta-acetic acid (DTPA) method and analyzed by Inductively Coupled Plasma Optical Emission Spectrometry (ICP-OES)." on page 3 lines 88-89 of the revised manuscript.*

19. Please put paragraph '2.3 Soil chemical analyses' before paragraph '2.2 Preparation of Fe oxide treatments';

**Re:** *We have revised it. Please see lines 83-89 and lines 90-105 in the revised manuscript.*

20. Paragraph '2.4 Experimental design and 15N addition': please specify the total N of samples. Is it 2x2x2x3=24x5 (time kinetics)=120?

**Re:** *Yes, the total experimental units is 120. We have revised it on lines 111-113 of the revised manuscript.*

21. Line 90: 'incubated for 6 days at 28 °C.'-> in the dark?

**Re:** *Yes, all the treatments were incubated in dark. We have revised it on line 111 of the revised manuscript.*

22. Lines 96-98: please specify in few words the techniques used for colorimetry and diffusion, and the model of machines;

**Re:** *The contents of $NH_4^+$ and $NO_3^-$ were quantified colorimetrically on a GENESYS 10 UV spectrophotometer (ThermoScientific, Madison, WI) using the salicylate method and the single reagent*

*method, respectively (Verdouw et al., 1978; Doane and Horwath, 2003). Isotope analysis of $NH_4^+$ and $NO_3^-$ was performed on aliquots of the extracts using a diffusion technique, by which $NH_4^+$ was distilled with Mg oxide and $NO_3^-$ was converted to $NH_4^+$ by Devarda's alloy and then distilled with Mg oxide. The $NH_3$ volatilizes were trapped using a boric acid solution (Feast and Dennis, 1996; Zhang et al., 2011, 2013). The $^{15}N$ isotopic composition in the trapped $NH_3$ volatilizes were then analyzed using an automated C/N analyzer coupled to an isotope ratio mass spectrometer (Europa Scientific Integra, UK). We have revised it on lines 119-126 of the revised manuscript.*

23. Line 100: 'MBN'. As for the other acronyms, once you defined them, use them always in the subsequent text;

**Re:** *We have revised it on lines 132, 179, 180, 413 and 442 of the revised manuscript.*

24. Line 104: 'were'->'was';

**Re:** *We have revised it on line 131 of the revised manuscript.*

25. Line 110: 'aprroximately'->' approximately';

**Re:** *We have revised it on line 137 of the revised manuscript.*

26. Paragraph '2.8 Statistical analyses': please state how you have checked the normality and homoscedasticity prior to ANOVA; please also specify the post-hoc tests you used;

**Re:** *We have revised it by "Differences in soil $NH_4^+$ and $NO_3^-$ content, net nitrification rate, gross mineralization rate, and MBN content among different treatments were assessed by analysis of variance (ANOVA). Prior to any statistical analysis, the normality of the data was evaluated by Shapiro–Wilk test and appropriate transformation (e.g. natural log-transformation) of the data was carried out if the transformation improved the normality. Post hoc Tukey's honestly significant difference multiple comparisons of means or paired t tests were used when appropriate to verify significant differences (P<0.05) between treatments. All statistical analyses were performed by SPSS statistical package." on lines 149-154 of the revised manuscript.*

RESULTS:

27. Lines 126-128: 'In both low and high pH soils, the $NH_4^+$-N concentrations showed a significant decrease after the application of $(NH_4)_2SO_4$, at both 30.9 and 15.6 mg $NH_4^+$-N $kg^{-1}$ soil at day 1 and 6, respectively, in the higher pH soil with the Fe oxide amendment' -> not clear, please rephrase;

**Re:** *We have revised this phrase by "The dynamics of soil inorganic N concentrations during the 6-day incubation are shown in Fig. 2. In both low and high pH soils with $(NH_4)_2SO_4$ application, the $NH_4^+$-N concentrations significantly decreased over the course of incubation. For example, in the high pH soil with Fe oxide and $(NH_4)_2SO_4$ were applied, the $NH_4^+$-N concentrations were 30.9 and 15.6 mg N $kg^{-1}$ soil at day 1 and 6, respectively (F=39.1, P=0.003)." on lines 157-160 of the revised manuscript.*

28. Lines 126-128: please describe the $+K^{15}NO_3$ figures and results;

**Re:** *We have added the results that "The $NO_3^--N$ concentrations increased significantly in all the $(NH_4)_2SO_4$ treatments during the incubation. However, the $NO_3^--N$ concentrations in all the $KNO_3$ treatments did not fluctuated significantly during the course of incubation (P > 0.05) (Fig. 2c and 2d)." Please see lines 160-162 in the revised manuscript.*

29. Lines 137-140: please state what is significant. . .;

**Re:** *We have added the significant sentence in the end "Compared with the control, the addition of Fe oxide significantly decreased the net nitrification rate in the high pH soil by 22.7 %, whereas 27.1% of net nitrification rate was increased by Fe oxide in the low pH soil (F = 63.1; P = 0.048) (Fig. 3b)." on lines 172-174 of the revised manuscript.*

30. Line 145: 'but slightly decreased it in the low pH soil'-> no, it is not significant so there is no decrease;

**Re:** *We have deleted it in the revised manuscript.*

DISCUSSION:
31. Line 158: 'suppressed'-> too strong. 'lowered'?;

**Re:** *We have deleted the word in the revised manuscript.*

32. Line 168: 'Kuroiwa et al., 2011'-> reference lacking;

**Re:** *We have added the information of "Kuroiwa, M., Koba, K., Isobe, K., Tateno, R., Nakanishi, A., Inagaki, Y., Toda, H., Otsuka, S., Senoo, K., Suwa, Y., Yoh, M., Urakawa, R. and Shibata, H.: Gross nitrification rates in four Japanese forest soils: heterotrophic versus autotrophic and the regulation factors for the nitrification, J. For. Res., 16, 363–373, 2011." in Reference section. Please see lines 287-289 in the revised manuscript.*

33. Line 169: 'it dominates nitrification'->' it generally dominates nitrification';

**Re:** *We have revised it on line 203 of the revised manuscript.*

34. Line 176: 'occurance'->' occurrence';

**Re:** *We have revised it on line 210 of the revised manuscript.*

35. Generally in the Discussion: please discuss the MBN15N results. . .; and discuss the impact on denitrification process;

**Re:** *We have revised it by "The addition of Fe oxide had no influence on MB$^{15}$N in the low pH soil, whereas in the high pH soil, 3.7 times higher MB$^{15}$N was found in the Fe oxide treatment than in the control (Fig. 4a). The high MB$^{15}$N content in the high pH soil with Fe oxide addition was probably related to the low activity of Fe oxide in the high pH soil due to the low solubility of Fe(III) oxide (Weber et al., 2006). Further research is needed to explore the mechanism of how the addition of Fe oxide increases microbial N assimilation in the high pH environment." on lines 225-230 in the revised manuscript.*

36. Line 183: 'Jansson et al., 1955'-> please find a more recent reference;

**Re:** *We have replenished the references by "(Jansson et al., 1955; Recous et al., 1990; Zhang et al., 2013)" and added the information of these references. Please see lines 217-218, lines 310-311 and lines 352-354 in the revised manuscript.*

TABLE 1:

37. Legend: 'studied soils' too vague, precise; precise what are fluvent/udifluvents subsamples. . . the two sites?

**Re:** *We hve changed "studied soils" to "the two soils with low pH and high pH". We have revised the table information, and the soil type of the two sites is fluvent/udifluvents. Please see lines 378-379 in the revised manuscript.*

38. The statistical data are lacking here! Specify as said above what is 'available Fe';

**Re:** *We have replenished the statistical data in the table on line 379. Avaibable Fe is the Fe which can be absorbed by soil microorganisms and plants and is extracted by DPTA (e.g. Wang, C., Ji, J., Yang, Z., Chen, L., Browne, P., and Yu, R.: Effects of soil properties on the transfer of cadmium from soil to wheat in the yangtze river delta region, China-a typical industry-agriculture transition area, Bological Trace Element Research, 148, 264-274, 2012).*

FIGURE 1:

39. Legend, line 357: 'moisture of'->'moisture at'; 'concentration'->'concentrations';

**Re:** *We have revised it on lines 408, 411, 414,417, 431, 437, 443 and 450 of the revised manuscript.*

FIGURE 2:

40. Legend: specify Fig 2a and Fig2b after mineralisation and nitrification;

**Re:** *We have added the information by "gross mineralization rate (a) and net nitrification rate (b)". Please see lines 410 and 436 in the revised manuscript.*

41. Are you sure that for Fig 2b pH5.1 control and pH 5.1 +Fe are statistically different?

**Re:** *We have checked the calculation process and statistical anaylsis results and come to the conclusion that there is a significant difference between pH 5.1 control and pH 5.1+Fe treatments (F = 63.1, P = 0.048).*

**FIGURE 3:**
42. Legend: specify acronyms + Fig 3a and Fig3b after 15N and N.

**Re:** *We have revised it by "MB$^{15}$N (a) and MBN (b)" on lines 413 and 442 of the revised manuscript.*

**List of all relevant changes made in the manuscript:**

**L10:** Replace "N transformations such as mineralization, immobilization, and nitrification depends on its redox activity, which can be…" with "N transformations including nitrification, mineralization, and immobilization, is influenced by redox activity, which is…"

**L11-12:** Replace "We hypothesized that the effect of Fe oxide on N transformation processes would be different in soils as a function of pH." with "The role of Fe minerals, particularly oxides, on affecting soil N transformation processes depends on soil pH, with Fe oxide often stimulating nitrification activity in the soil with low pH."

**L12-14:** Replace "This study aimed to investigate N mineralization-immobilization, especially nitrification, as affected by Fe oxide in soils with different pH. A set of lab incubations under 100 % water holding capacity were carried out to investigate the effect of Fe oxide on N transformation rates in two subtropical agricultural soils with a low pH (pH 5.1) and a high pH (pH 7.8)." with "We conducted lab incubations to investigate the effect of Fe oxide on N transformation rates in two subtropical agricultural soils with low pH (pH 5.1) and high pH (pH 7.8)."

**L15-16:** Replace "Iron oxide addition stimulated net nitrification in the low pH soil (pH 5.1), while the opposite occurred in the high pH soil (pH 7.8)." with "Iron oxide stimulated net nitrification in low pH soil (pH 5.1), while the opposite occurred in high pH soil (pH 7.8)."

**L16-17:** Add "Compared to the control, Fe oxide decreased microbial immobilization of inorganic N by 50% in low pH soil but increased it by 45% in high pH soil."

**L17-18:** Replace "An explanation for this could be at low pH, Fe oxide increased $NH_3$-N availability by stimulating N mineralization and inhibiting N immobilization." with "A likely explanation for the effects at low pH is that Fe oxide increased $NH_3$-N availability by stimulating N mineralization and inhibiting N immobilization."

**L18-20:** Replace "These results suggested that Fe oxide plays an important role in N transformations in soil ecosystem, and the effect of Fe oxide on N transformations depends on soil pH." with "These results indicate that Fe oxide plays an important role in soil N transformation processes and the magnitude of the effect of Fe oxide is depending significantly on soil pH."

**L24:** Add "The effect of soil pH on redox sensitive of soil N transformations, especially nitrification, is receiving increasing attention."

**L25:** Replace "the" with "a", Replace "-3 ($NH_4^+$)" with "–3 (ammonium: $NH_4^+$)".

**L26-27:** Replace "($NO_3^-$), including compounds with intermediate oxidation states such as $NH_2OH$ and $NO_2^-$ which are formed to various degrees during nitrification;" with "(nitrate: $NO_3^-$), including compounds with intermediary oxidation states such as hydroxylamine ($NH_2OH$) and nitrite ($NO_2^-$) which are formed to various degrees during nitrification.".

**L27-28:** Replace "it is therefore a key process in the global N cycle" with "This process also can produce reactive intermediates such as nitrogen oxides (NOx) and nitrous oxide ($N_2O$) affecting air quality and global climate.".

**L28-29:** Replace "and its oxides affect…" with "oxides on…".

**L30:** Replace "et al., 2015a), yet its influence is rarely identified in biogeochemical cycles and models." with "et al., 2015a) is recognized, yet is rarely identified in biogeochemical models that predict global N cycle processes."

**L30-33:** Add "Iron and its oxides are found in abundance in many soils, with large amounts of Fe oxides are typically found in subtropical and tropical soils. Thus, understanding of the relationship between Fe oxides and soil nitrification process is especially important for understanding its influence on N cycling processes."

**L34:** Add "Li et al., 1988;"

**L35-36:** Replace "referring to anaerobic ammonium ($NH_4^+$) oxidation coupled to Fe(III) reduction resulting in nitrate ($NO_3^-$) (Luther et al., 1997) or nitrite ($NO_2^-$) (Clément et al., 2005; Sawayama, 2006; Shrestha et al., 2009). This process" with "referred to as anaerobic $NH_4^+$ oxidation coupled to Fe(III) reduction. Nitrate and dinitrogen ($N_2$) are produced during this process (Luther et al., 1997; Clément et al., 2005; Sawayama, 2006; Shrestha et al., 2009; Yang et al., 2012). Feammox"

**L37:** Replace "wetland or saturated soils…" with "saturated soils such as wetland…"

**L38:** Replace "may play a critical role as an electron acceptor" with "can act as an electron acceptor and play a critical role influencing N reactions"

**L40:** Replace "is also involved…" with "can also participate…"

**L42-43:** Add "In ten agricultural soils, Zhu et al. (2013) found that Fe ranked higher than any other intrinsic soil property in affecting nitrous oxide ($N_2O$) production and emission."

**L43-44:** Replace "Oxidation of Fe (II) coupled to $NO_3^-$ reduction can proceed via biotic and abiotic pathways in wetland soils, sediments or anoxic microsites…" with "In wetland soils, sediments or anoxic microsites, oxidation of Fe(II) coupled to $NO_3^-$ reduction proceed simultaneously via biotic and abiotic pathways…"

**L45-47:** Replace "The coupling between biotic (e.g. Fe(III)-reducing microoganisms) and abiotic factors (e.g. pH) mediates the redox cycle of Fe that can lead to organic matter decomposition and thus N mineralization under oxic or anoxic conditions…" with "Under oxic or anoxic conditions, the interaction between biotic (e.g. Fe(III)-reducing microorganisms) and abiotic factors (e.g. pH) mediates the redox cycle of Fe, which can lead to organic matter decomposition and thus N mineralization…"

**L49:** Replace "are assumed to via…" with "and is assumed to occur via…"

**L50:** Replace "releasing the dissolved organic carbon that complexed…" with "the release of dissolved organic carbon that can complex…"

**L50-51:** Replace "Previous study with…" with "A study on…"

**L52:** Replace "Fe (III), suggesting that Fe-organic matter compound is an important factor that influences…" with "Fe(III). It suggests that Fe-organic matter complexes are important in influencing…"

**L54:** Replace "can also…" and "small…" with "can…" and "a small…", respectively.

**L56:** Replace "was" with "were"

**L56-58:** Add "Studies on Fe requirements for ammonia-oxidizing bacterial (AOB) showed that when the Fe concentration in the medium of *Nitrosomonas europaea* culture increased from 0.2 to 10 µM Fe, the activities of both ammonia monooxygenase and hydroxylamine oxidoreductase decreased (Wei et al., 2006)."

**L59:** Replace "showed" and "ammonia-oxidizing bacterial (AOB)" with "observed" and "AOB", respectively. Delete "were".

**L61:** Replace "iron oxide addition in different pH soils" with "Fe oxide addition under varying soil pH." Replace "since…" with "This is because…"

**L64:** Replace "transformation processes would differ depending on…" with "transformations depends on a large extent on…"

**L66:** Replace "different" with "varying"

**L66-67:** Replace "How does Fe oxide influence these N transformation processes under different pH in soils with 100 % water holding capacity (WHC)?" with "What is the mechanism of Fe oxide that influenced N transformations under different soil pH?"

**L67:** Replace "effects" with "affects"

**L72:** Replace "days." with "days per year."

**L72-73:** Replace "Soil samples were collected from agricultural land (29.70° N, 106.38° E) with low pH soil (pH 5.1) and a hill site (29.75° N, 106.40° E) with high pH soil (pH 7.8)." with "Soil samples were collected from agricultural land (29.70° N, 106.38° E) with low pH soil (pH 5.1) and a hill site (29.75° N, 106.40° E) with high pH soil (pH 7.8) in March, 2015."

**L73-74:** Replace "The soils were derived…" with "Both soils were developed…"

**L75:** Replace "…sample was collected…" with "…was sampled…"

**L75-77:** Replace "…sweet potato." with "…sweet potato under conventional cultivation over ten years. In spring maize and autumn sweet potato growing seasons, N fertilizers were conventionally applied as urea at rates of 75 and 225 kg N ha$^{-1}$, respectively."

**L77:** Replace "The high pH soil sample was collected from pear orchard, which was converted from farmland a few years ago and never been fertilized and tilled since then." with "The high pH soil was sampled from a pear orchard, which was converted from cropland three years ago and never been fertilized or tilled since the conversion."

**L81:** Replace "a moisture" and "to use" with "a gravimentric moisture" and "to use (within two months)", respectively.

**L82:** Replace "analyses" with "analyses (Jiang et al., 2015a)"

**L83:** Change the order between section 2.2 and 2.3.

**L84:** Add "The results for the soil chemical properties are shown in Table 1."

**L86-89:** Replace "Total soil Fe and free Fe oxides were extracted with HNO$_3$-HF-HClO$_4$ and Na$_2$S$_2$O$_4$-Na$_3$C$_6$H$_5$O$_7$-NaHCO$_3$, respectively. The concentration of Fe was measured by atomic absorption spectrophotometry with a graphite furnace (GFAAS) using a model Z-8200 spectrophotometer" with "Total soil Fe was extracted with HNO$_3$-HF-HClO$_4$ and measured by atomic absorption spectrophotometry with a graphite furnace (GFAAS) using a model Z-8200 spectrophotometer. Available Fe was extracted using the diethylenetriamine penta-acetic acid (DTPA) method and analyzed by Inductively Coupled Plasma Optical Emission Spectrometry (ICP-OES)"

**L91:** Replace "preparation…" with "preparation in this study…"

**L92:** Replace "A mixture of iron (III)…" with "Briefly, Fe(III)…"

**L93-94:** Replace "stirred, followed by pH adjustment to 7–8 with 1 mol L$^{-1}$ KOH and then…" with "first mixed and stirred. Then, the pH of mixture was adjusted to 7–8 with 1 mol L$^{-1}$ KOH and…"

**L95:** Replace "The particle density of ferrihydrite in the final suspension was 87 g L$^{-1}$" with "The particle density and original pH of ferrihydrite in the final suspension were 87 g L$^{-1}$ and 3.7, respectively"

**L96-97:** Replace "freeze-dried to analyze for ferrihydrite using the X-ray diffractometer (XRD)" with "freeze-dried and then analyzed for ferrihydrite using the X-ray diffractometer (XRD) (PANalytical B.V., Holland) (Fig. 1)"

**L97-99:** Replace "The remaining ferrihydrite suspension was divided and adjusted to either pH of 5.1 or 7.8 to match the initial pH in the different soils." with "The remaining ferrihydrite suspension was divided into two parts. The pH of these two parts were adjusted to either 5.1 or 7.8, the same as the initial pH in low pH or high pH soils."

**L100-101:** Replace "amended with Fe (adjusted the pH of ferrihydrite to match the soil pH)" with "Fe treated (the pH of ferrihydrite was adjusted to the same as soil pH)"

**L101:** Replace "the ferrihydrite" with "Fe oxide"

**L103:** Replace "added" with "added according to the treatment design"

**L104-105:** Replace "soils, then the soil mixtures were slightly air-dried to reach a moisture content of about 15 %, passed through a sieve of 2 mm and stored at 4 °C before use" with "soils. The soil mixtures were then slightly air-dried to reach a gravimentric moisture content of about 15 %, and passed through a sieve of 2 mm and stored at 4 °C before use (within 7 days)"

**L106:** Replace "Experimental design and $^{15}$N addition" with "Soil incubation with $^{15}$N substrates"

**L108:** Replace "20 g (dry mass) Fe treated or non-Fe treated soils were weighed" with "Fe treated or non-Fe treated soils were weighed (20 g dry mass)"

**L110-111:** Replace "Soils in all the treatments were adjusted to 100 % WHC and incubated for 6 days at 28 °C" with "All treatments were incubated at 100 % water holding capacity (WHC) at 28 °C for 6 days in dark after N application"

**L111-114:** Add "The experimental design and treatment application was set as a completely randomized block design, with three replicates per treatment (120 total experimental units comprising 5 soil sampling times). 100% WHC was chosen to create an oxic-anoxic interface, in which the redox cycle of Fe oxide commonly exists."

**L117:** Replace "soils (three replicates for each treatment)" with "three replicates for each treatment"

**L118:** Replace "after the application of N" with "after N application"

**L119-121:** Replace "analyzed using colorimetric methods…" with "quantified colorimetrically on a GENESYS 10 UV spectrophotometer (ThermoScientific, Madison, WI) using the salicylate method and the single reagent method, respectively…"

**L122-124:** Replace "technique…" with "technique, by which $NH_4^+$ was distilled with Mg oxide and $NO_3^-$ was converted to $NH_4^+$ by Devarda's alloy and then distilled with Mg oxide. The $NH_3$ volatilizes were trapped using a boric acid solution…"

**L124-125:** Replace "$NH_4^+$-N and $NO_3^-$-N were analyzed…" with "the trapped $NH_3$ volatilizes were then

analyzed…"

**L127-128:** Replace "For soil microbial biomass N (MBN) analysis, soils were fumigated or unfumigated at time 0 and 6 days after N application, then extracted with 0.5 mol $L^{-1}$ $K_2SO_4$ (5 to 1 extractant volume to soil mass ratio)…" with "Soils were fumigated or unfumigated then extracted with 0.5 mol $L^{-1}$ $K_2SO_4$ (5 to 1 extractant volume to soil mass ratio) at time 0 and 6 days after N application for soil microbial biomass N (MBN) determination…"

**L129:** Replace "was stored" with "stored"

**L131:** Replace "were" with "was"

**L132:** Replace "Microbial biomass N" with "MBN"

**L137:** Replace "aprroximately" with "approximately"

**L142:** Replace "with the equation described by" with "according to"

**L143:** Replace "of" with "in"

**L149-153:** Replace "Data were subjected to one-way ANOVA and mean values were separated using Duncan's New Multiple Range Test at $P$ < 0.05." with "Differences in soil $NH_4^+$ and $NO_3^-$ content, net nitrification rate, gross mineralization rate, and MBN content among different treatments were assessed by analysis of variance (ANOVA). Prior to any statistical analysis, the normality of the data was evaluated by Shapiro–Wilk test and appropriate transformation (e.g. natural log-transformation) of the data was carried out if the transformation improved the normality. Post hoc Tukey's honestly significant difference multiple comparisons of means or paired t tests were used when appropriate to verify significant differences (P<0.05) between treatments."

**L157:** Replace "1" with "2"

**L158-160:** Replace "soils, the $NH_4^+$-N concentrations showed a significant decrease after the application of $(NH_4)_2SO_4$, at both 30.9 and 15.6 mg $NH_4^+$-N $kg^{-1}$ soil at day 1 and 6, respectively, in the higher pH soil with the Fe oxide amendment." with "soils with $(NH_4)_2SO_4$ application, the $NH_4^+$-N concentrations significantly decreased over the course of incubation. For example, in the high pH soil with Fe oxide and $(NH_4)_2SO_4$ were applied, the $NH_4^+$-N concentrations were 30.9 and 15.6 mg N $kg^{-1}$ soil at day 1 and 6, respectively (F=39.1, P=0.003)."

**L161-162:** Add "However, the $NO_3^-$-N concentrations in all the $KNO_3$ treatments did not fluctuated significantly during the course of incubation ($P$ > 0.05) (Fig. 2c and 2d)."

**L164:** Replace "Fe oxide amended" with "Fe treated"

**L165:** Replace "phenomenon was found" with "occurred", Replace "2" with "3"

**L166:** Replace "Fe oxide" and "were" with "Fe oxide amendment" and "was", respectively.

**L167:** Replace "Fe oxide" and with "Fe oxide amendment"

**L168-169:** Replace "with the Fe oxide, but the non-Fe oxide (control) low pH soil had significantly lower gross N mineralization than the high pH soil" with "with Fe oxide amendment, but the non-Fe treated (control) high pH soil had the highest gross N mineralization among all the treatments"

**L170:** Replace "the Fe oxide" with "Fe oxide amendment"

**L171-172:** Replace "the Fe oxide" with "Fe oxide amendment"

**L172-173:** Replace "The application of Fe oxide decreased the net nitrification rate in the high pH soil by 22.7 %, whereas the net nitrification rate in the low pH soil was increased 27.1 % with Fe oxide (Fig. 2b)" with "Compared with the control, the addition of Fe oxide significantly decreased the net nitrification rate in the high pH soil by 22.7%, whereas 27.1% of net nitrification rate was increased by Fe oxide in the low pH soil (F = 63.1; P = 0.048) (Fig. 3b)"

**L177-178:** Replace "3", "the microbial biomass (MB$^{15}$N)" and "Fe oxide" with "4", "MB$^{15}$N" and "Fe oxide amendment", respectively.

**L179:** Replace "MB$^{15}$N but slightly decreased it…" with "MB$^{15}$N…"

**L180:** Replace "3" and "the microbial biomass (MBN)" with "4" and "MBN", respectively.

**L181:** Replace "Fe oxide" with "Fe oxide amendment"

**L182:** Replace "soil with Fe oxide" with "soil"

**L183:** Replace "3" with "4"

**L183-184:** Add "Compared with control, Fe oxide addition decreased MBN by 50% in the low pH soil but increased it by 45% in the high pH soil."

**L186:** Replace "Fe oxide" with "Fe oxide amendment"

**L187:** Replace "4" with "5"

**L188-189:** Replace "soil, the concentration of Fe (II) did not change between day 0 and days 6 after the addition of Fe oxide" with "soil amended with Fe oxide, the concentration of Fe(II) did not change between day 0 and days 6"

**L191-192:** Replace "The effect of Fe oxide on the nitrification rate varied in different pH soils under 100 %

WHC, supporting our hypothesis in this study." with "In the present study, the addition of Fe oxide increased the net nitrification rate by 20.8 % in the low pH soil. The effect of Fe oxide on the nitrification rate varied with soil pH, supporting our hypothesis."

**L192-196:** Add "Nitrification is primarily dependent on $NH_3$ availability and the activity of nitrifying microorganisms. A regression-analysis assessing relationships among the rates of N transformation processes from 100 published studies with nearly 300 different organic and mineral soil materials concluded that nitrification rate is controlled by the rate of ammonia released from soil organic matter mineralization (Booth et al. 2005)."

**L196-197:** Replace "The addition of Fe oxide stimulated the net nitrification rate in the low pH soil (pH 5.1) (F = 63.13; P = 0.048), but suppressed…" with "The addition of Fe oxide had opposite effects on nitrification, stimulating it at low pH soil (pH 5.1) (F = 63.13; P = 0.048) and lowering…"

**L198:** Replace "Nitrification is primarily dependent on $NH_3$ availability and the activities of nitrifying microorganisms. Factors that affect $NH_3$ availability and nitrifying microorganisms can directly influence the nitrification process." with "The amount of substrate ammonia available for nitrification is dependent on the gross N mineralization rate."

**L199-200:** Replace "The increased N mineralization $R–NH_2 \rightarrow NH_3$ rate at low pH increased the substrate availability for nitrification in presence of Fe oxide." with "An increase in N mineralization ($R–NH_2 \rightarrow NH_3$) likely increased the availability of $NH_3$, leading to an increase in nitrification."

**L203:** Replace "it dominates" with "it generally dominates"

**L207:** Replace "The significant increase in the Fe (II)" with "This assumption was supported by the significant increase in Fe(II)"

**L208:** Replace "Fe oxide supported this assumption (Fig. 4)" and "is higher" with "Fe oxide amendment (Fig. 5)" and "is generally high", respectively.

**L209:** Replace "condition in our incubation conditions" with "soil moisture in our incubation"

**L210:** Replace "occurance" with "occurrence"

**L211:** Replace "explain the increase of the net nitrification rate in the low pH" with "explain exclusively the increase in the net nitrification rate in low pH"

**L213-214:** Replace "rate. Immobilization of inorganic N increased due to the addition of Fe oxide in the high pH soil, might have contributed to the decrease in net nitrification (Fig. 3b)." with "rate, likely due to increased inorganic N immobilization (Fig. 4b)."

**L214-216:** Replace "could be an important reason for the decrease in nitrification in the high pH soil. Meiklejohn (1953) indicated that high concentrations of Fe (> 112 mg $L^{-1}$)…" with "can be another

important reason for Fe oxide decreasing nitrification at high pH. For example, Meiklejohn (1953) demonstrated that high Fe (> 112 mg $L^{-1}$)…"

**L218:** Add "Recous et al., 1990; Zhang et al., 2013"

**L219:** Replace "suggesting the microbial biomass maybe flexible in utilizing N sources (Stark and Hart, 1997)" with "(Stark and Hart, 1997), suggesting that microbial biomass maybe flexible in utilizing different N sources."

**L220:** Replace "that affects" with "affecting"

**L221:** Replace "Fe oxide addition" with "the addition of Fe oxide"

**L222-223:** Replace "(Fig. 3b). This could indicate that the high solubility of Fe oxide at low pH could impair the assimilation of N by…" with "(Fig. 4b). This indicates that high solubility of Fe oxide at low pH environment can impair the assimilation of N and reduce the size of…"

**L224:** Replace "could" with "can"

**L225:** Replace "lead to N mineralization and ammonification, thus increasing…" with "lead to an increase in N mineralization and ammonification, thus increase…"

**L225-227:** Add "The addition of Fe oxide had no influence on MB$^{15}$N in the low pH soil, whereas in the high pH soil, 3.7 times higher MB$^{15}$N was found in the Fe oxide treatment than in the control (Fig. 4a)."

**L227-228:** Replace "Compared with the low pH soil, the activity of Fe oxide in the high pH soil was low…" with "The high MB$^{15}$N content in the high pH soil with Fe oxide addition was probably related to the low activity of Fe oxide in the high pH soil…"

**L229-230:** Replace "Thus, no or low Fe(III) reduction likely occurred in the high pH soil" with "Further research is needed to explore the mechanism of how the addition of Fe oxide increases microbial N assimilation in the high pH environment"

**L233:** Delete "the"

**L234:** Replace "according to" with "with"

**L235:** Replace "elucidate mechanisms on…" with "develop the mechanistic understanding of…"

**L239:** Add "Xia Zhu-Barker revised and edited the manuscript."

**L240:** Replace "discussion" with "writing"

**L287-291:** Add "Kuroiwa, M., Koba, K., Isobe, K., Tateno, R., Nakanishi, A., Inagaki, Y., Toda, H.,

Otsuka, S., Senoo, K., Suwa, Y., Yoh, M., Urakawa, R., and Shibata, H.: Gross nitrification rates in four Japanese forest soils: heterotrophic versus autotrophic and the regulation factors for the nitrification, J. For. Res., 16, 363–373, 2011.

Li, L., Pan, Y., Wu, Q., Zhou, X., and Li, Z.: Investigation on Amorphous ferric oxide acting as an electron acceptor in the oxidation of $NH_4^+$ under anaerobic condition, Acta Pedologica Sinica, 25, 184-190, 1988."

**L310-311:** Add "Recous, S., Mary, B., and Faurie, G.: Microbial immobilization of ammonium and nitrate in cultivated soils, Soil Biol. Biochem., 22, 913–922, 1990."

**L342-343:** Add "Wei, X., Vajrala, N., Hauser, L., Sayavedra-Soto, L. A., and Arp, D. J.: Iron nutrition and physiological responses to iron stress in Nitrosomonas europaea, Arch. Microbiol., 186, 107–118, 2006."

**L352-356:** Add "Zhang, J. B., Zhu, T. B., Meng, T. Z., Zhang, Y. C., Yang, J. J., Yang, W. Y., Müller, C., and Cai, Z. C.: Agricultural land use affects nitrate production and conservation in humid subtropical soils in China, Soil Biol. Biochem., 62, 107–114, 2013.

Zhu, X., Silva, L. C., Doane, T. A., and Horwath, W. R.: Iron: the forgotten driver of nitrous oxide production in agricultural soil, Plos One 8, e60146, 2013."

**L378:** Replace "studied soils" with "the soils with low pH and high pH"

**L379:** Replace "mg N $kg^{-1}$ $d^{-1}$" with "mg N $kg^{-1}$".

Replace

| Subsamples | pH | Organic matter g $kg^{-1}$ | Total N g $kg^{-1}$ | Total Fe g $kg^{-1}$ | Available Fe mg $kg^{-1}$ | $NO_3^-$-N mg N $kg^{-1}$ $d^{-1}$ | $NH_4^+$-N mg N $kg^{-1}$ $d^{-1}$ |
|---|---|---|---|---|---|---|---|
| Fluvents Udifluvents | 5.1 | 18.0 | 0.73 | 16.3 | 132 | 10.3 a | 1.54 b |
| | 7.8 | 13.9 | 0.68 | 27.5 | 5.64 | 4.68 b | 2.44 a |

with

| Soil type | pH | Organic matter g $kg^{-1}$ | Total N g $kg^{-1}$ | Total Fe g $kg^{-1}$ | Available Fe mg $kg^{-1}$ | $NO_3^-$-N mg N $kg^{-1}$ | $NH_4^+$-N mg N $kg^{-1}$ |
|---|---|---|---|---|---|---|---|
| Fluvents Udifluvents | 5.1 | $18.0 \pm 0.26$ a | $0.73 \pm 0.01$ a | $16.3 \pm 0.08$ b | $132 \pm 4.04$ a | $10.3 \pm 0.85$ a | $1.54 \pm 0.19$ b |
| Fluvents Udifluvents | 7.8 | $13.9 \pm 0.11$ b | $0.68 \pm 0.03$ a | $27.5 \pm 0.04$ a | $5.64 \pm 0.49$ b | $4.68 \pm 0.48$ b | $2.44 \pm 0.16$ a |

**L406:** Add "**Figure 1:** X-ray diffraction pattern of ferrihydrite."

**L407:** Replace "1" with "2"

**L408:** Replace "of" and "concentration" with "at" and "concentrations", respectively.

**L410:** Replace "**2:** Effects of Fe oxide on gross mineralization rate and net nitrification rate…" with "**3:** Effects of Fe oxide on gross mineralization rate (a) and net nitrification rate (b)…"

**L411:** Replace "of" with "at"

**L413:** Replace "**3:** Effects of Fe oxide on microbial biomass $^{15}$N and microbial biomass N…" with "**4:** Effects of Fe oxide on MB$^{15}$N (a) and MBN (b)…"

**L414:** Replace "of" with "at"

**L416:** Replace "4" with "5"

**L417:** Replace "of" with "at"

**L424-425:** Add

[Figure]

**Figure 1:** X-ray diffraction pattern of Ferrihydrite.

**L430:** Replace "1" with "2"

**L431:** Replace "of" and "concentration" with "at" and "concentrations", respectively.

**L436:** Replace "**2:** Effects of Fe oxide on gross mineralization rate and net nitrification rate…" with "**3:** Effects of Fe oxide on gross mineralization rate (a) and net nitrification rate (b)…"

**L437:** Replace "of" with "at"

**L442:** Replace "**3:** Effects of Fe oxide on microbial biomass $^{15}$N and microbial biomass N…" with "**4:** Effects of Fe oxide on MB$^{15}$N (a) and MBN (b)…"

**L443:** Replace "of" with "at"

**L449:** Replace "4" with "5"

**L450:** Replace "of" with "at"

[revised manuscript text omitted]
 | 5.1 | $18.0 \pm 0.26$ a | $0.73 \pm 0.01$ a | $16.3 \pm 0.08$ b | $132 \pm 4.04$ a | $10.3 \pm 0.85$ a | $1.54 \pm 0.19$ b |
| Fluvents Udifluvents | 7.8 | $13.9 \pm 0.11$ b | $0.68 \pm 0.03$ a | $27.5 \pm 0.04$ a | $5.64 \pm 0.49$ b | $4.68 \pm 0.48$ b | $2.44 \pm 0.16$ a |

1280

1285

1290

1295

**Figure Captions**

**Figure 1:** X-ray diffraction pattern of ferrihydrite.

**Figure 2:** Effects of Fe oxide on $NH_4^+$-N and $NO_3^-$-N dynamics during 6-day by $^{15}N$ tracing incubation at 28 ºC with soil moisture at 100 % WHC. $NH_4^+$-N and $NO_3^-$-N concentrations were measured following the addition of 50 mg N kg$^{-1}$ ($^{15}NH_4$)$_2$SO$_4$ (a and b) and K$^{15}NO_3$ (c and d). Error bars represent standard deviation, n = 3.

**Figure 2:** Effects of Fe oxide on gross mineralization rate (a) and net nitrification rate (b) during 6-day for incubated soil samples incubation at 28 ºC with soil moisture at 100 % WHC. Error bars represent standard deviations, n = 3. The different letters above the columns indicate a significant difference ($P < 0.05$).

**Figure 3:** Effects of Fe oxide on  MB$^{15}$N (a) and  MBN (b) pools during 6-day with ($^{15}NH_4$)$_2$SO$_4$ treatment incubation at 28 ºC with soil moisture at 100 % WHC. Error bars represent standard deviations, n = 3. The different letters above the columns indicate a significant difference ($P < 0.05$).

**Figure 4:** Effects of Fe oxide on concentration of Fe(II) (0.5 mol L$^{-1}$ HCl extractable) before and after 6-day with ($^{15}NH_4$)$_2$SO$_4$ treatment incubation at 28 ºC with soil moisture at 100 % WHC. Error bars represent standard deviations, n = 3. The different letters above the columns indicate a significant difference ($P < 0.05$).

[Figure]

**Figure 1:** X-ray diffraction pattern of Ferrihydrite.

1320

[Figure]

**Figure 2:** Effects of Fe oxide on $NH_4^+$-N and $NO_3^-$-N dynamics during 6-day by $^{15}N$ tracing incubation at 28 ºC with soil moisture at 100 % WHC. $NH_4^+$-N and $NO_3^-$-N concentration were measured following the addition of 50 mg N kg$^{-1}$ ($^{15}NH_4$)$_2$SO$_4$ (a and b) and K$^{15}NO_3$ (c and d). Error bars represent standard deviation, n = 3.

1325

[Figure]

**Figure 3**: Effects of Fe oxide on gross mineralization rate (a) and net nitrification rate (b) during 6-day for incubated soil samples incubation at 28 ℃ with soil moisture at 100 % WHC. Error bars represent standard deviations, n = 3. The different letters above the columns indicate a significant difference ($P < 0.05$).

[Figure]

**Figure** 4: Effects of Fe oxide on  MB$^{15}$N (a) and  MBN (b) pools during 6-day with ($^{15}$NH$_4$)$_2$SO$_4$ treatment incubation at 28 ºC with soil moisture of 100 % WHC. Error bars represent standard deviations, n = 3. The different letters above the columns indicate a significant difference ($P < 0.05$).

[Figure]

1340

**Figure 45:** Effects of Fe oxide on concentration of Fe(II) (0.5 mol L$^{-1}$ HCl extractable) before and after 6-day with ($^{15}$NH$_4$)$_2$SO$_4$ treatment incubation at 28 ℃ with soil moisture at 100 % WHC. Error bars represent standard deviations, n = 3. The different letters above the columns indicate a significant difference ($P < 0.05$).

---

## Referee Report (RR1)

[referee-annotated manuscript omitted]